# KAGNNs: Kolmogorov-Arnold Networks meet Graph Learning

**Roman Bresson**
*KTH Royal Institute of Technology, Sweden*                                                      *bresson@kth.se*

**Giannis Nikolentzos**
*University of Peloponnese, Greece*                                                            *nikolentzos@uop.gr*

**George Panagopoulos**
*University of Luxembourg, Luxembourg*                                                *georgios.panagopoulos@uni.lu*

**Michail Chatzianastasis**
*École Polytechnique, IP Paris, France*                                        *michail.chatzianastasis@polytechnique.edu*

**Jun Pang**
*University of Luxembourg, Luxembourg*                                                         *jun.pang@uni.lu*

**Michalis Vazirgiannis**
*École Polytechnique, IP Paris, France*
*KTH Royal Institute of Technology, Sweden*                                             *mvazirg@lix.polytechnique.fr*

**Reviewed on OpenReview:** *https://openreview.net/forum?id=O3UB1MCAMr*

## Abstract

In recent years, Graph Neural Networks (GNNs) have become the de facto tool for learning node and graph representations. Most GNNs typically consist of a sequence of neighborhood aggregation (a.k.a., message-passing) layers, within which the representation of each node is updated based on those of its neighbors. The most expressive message-passing GNNs can be obtained through the use of the sum aggregator and of MLPs for feature transformation, thanks to their universal approximation capabilities. However, the limitations of MLPs recently motivated the introduction of another family of universal approximators, called Kolmogorov-Arnold Networks (KANs) which rely on a different representation theorem. In this work, we compare the performance of KANs against that of MLPs on graph learning tasks. We implement three new KAN-based GNN layers, inspired respectively by the GCN, GAT and GIN layers. We evaluate two different implementations of KANs using two distinct base families of functions, namely B-splines and radial basis functions. We perform extensive experiments on node classification, link prediction, graph classification and graph regression datasets. Our results indicate that KANs are on-par with or better than MLPs on all tasks studied in this paper. We also show that the size and training speed of RBF-based KANs is only marginally higher than for MLPs, making them viable alternatives. Code available at https://github.com/RomanBresson/KAGNN.

# 1 Introduction

Graphs are structural representations of information useful for modeling different types of data. They arise naturally in a wide range of application domains, and their abstract nature offers increased flexibility. Typically, the nodes of a graph represent entities, while the edges capture the interactions between them. For instance, in social networks, nodes represent individuals, and edges represent their social interactions. In chemo-informatics, molecules are commonly modeled as graphs, with nodes corresponding to atoms and edges to chemical bonds. In other settings, molecules can also correspond to nodes, with edges capturing their ability to bond with one another.

In many cases where graph data is available, there exist problems that cannot be solved efficiently using conventional tools (e. g., graph algorithms) and require the use of machine learning techniques. For instance, in the field of chemo-informatics, the standard approach for estimating the quantum mechanical properties of molecules leverages computationally expensive density functional theory computations (Gilmer et al., 2017). Machine learning methods could serve as a more efficient alternative to those methods. Recently, Graph Neural Networks (GNNs) have been established as the dominant approach for learning on graphs (Wu et al., 2020). Most GNNs consist of a series of message-passing layers. Within a message-passing layer, each node updates its feature vector by aggregating the feature vectors of its neighbors and combining the emerging vector with its own representation.

A lot of recent work has focused on investigating the expressive power of GNNs (Li and Leskovec, 2022). There exist different definitions of expressive power, however, the most common definition is concerned with the number of pairs of non-isomorphic graphs that a GNN model can distinguish. Two graphs are isomorphic if there exists an edge-preserving bijection between their respective sets of nodes. In this setting, a model is more expressive than another model if the former can distinguish all pairs of non-isomorphic graphs that the latter can distinguish, along with other pairs that the latter cannot (Morris et al., 2019). Furthermore, an equivalence was established between the ability of GNNs to distinguish non-isomorphic graphs and their ability to approximate permutation-invariant functions on graphs (Chen et al., 2019). This line of work gave insights into the limitations of different models (Xu et al., 2019; Nikolentzos et al., 2020), but also led to the development of more powerful architectures (Morris et al., 2019; Maron et al., 2019; Morris et al., 2020a).

Most maximally-expressive GNN models rely on multi-layer perceptrons (MLPs) as their main building blocks, due to their universal approximation capabilities (Cybenko, 1989; Hornik et al., 1989). The theorem states that any continuous function can be approximated by an MLP with at least one hidden layer, given that this layer contains enough neurons. Recently, Kolmogorov-Arnold Networks (KANs) (Liu et al., 2025) have emerged as promising alternatives to MLPs. They are based on the Kolmogorov-Arnold representation theorem (Kolmogorov, 1957) which states that a continuous multivariate function can be represented by a composition and sum of a fixed number of univariate functions. KANs substitute the learnable weights and pre-defined activation functions of MLPs with learnable activations and summations. The initial results demonstrate that KANs have the potential to be more accurate than MLPs in low dimensions and in settings where regularity is expected.

In this paper, we present a thorough empirical comparison of the performance between GNNs that use KANs to update node representations and GNNs that utilize MLPs to that end. Our intuition is that KANs could exploit the expected regularity underlying the message-passing paradigm to provide comparatively good performance on graph tasks. Our work is orthogonal to prior work that studies the expressive power of GNNs since we empirically compare models that are theoretically equally expressive in terms of distinguishing non-isomorphic graphs against each other, and we study the impact of the different function approximation modules (i. e., KANs or MLPs) on the models' performance. We implement two layers, one based on the Graph Convolutional Network (GCN) (Kipf and Welling, 2017) and one based on the Graph Isomorphism Network (GIN) (Xu et al., 2019). We evaluate the different GNN models on several standard node classification, link prediction, graph classification, and graph regression datasets.

The rest of this paper is organized as follows. Section 3, provides an overview of the tasks we address in this paper, as well as a description of message-passing GNNs and Kolmogorov-Arnold networks. In Section 4, we introduce the `KAGIN` (Kolmogorov-Arnold Graph Isomorphism Network), `KAGCN` (Kolmogorov-Arnold

Graph Convolution Network) and `KAGAT` (Kolmogorov-Arnold Graph ATtention Network) models, which are variants of existing GNNs, and which leverage KANs to update node features within each layer. In Section 5, we present extensive empirical results comparing the above models with their vanilla counterparts in several tasks, and in Section 6 we evaluate the time efficiency of our architectures. Finally, Section 7 presents perspectives for future work and Section 8 concludes the paper.

## 2    Related Work

Kolmogorov-Arnold Networks have recently gained significant attention in the machine learning community (Somvanshi et al., 2024). Even though KANs were introduced very recently, they have already been applied to different problems such as in the task of satellite image classification (Cheon, 2024), for predicting the pressure and flow rate of flexible electrohydrodynamic pumps (Peng et al., 2024), and for rumor detection (Jiang et al., 2024). So far, most efforts have focused on time series data (Xu et al., 2024a; Barašin et al., 2024). For instance, KANs have been evaluated in the satellite traffic forecasting task (Vaca-Rubio et al., 2024). Furthermore, they were combined with architectures usually leveraged in time series forecasting tasks such as LSTM networks (Genet and Inzirillo, 2024a) and the Transformer (Genet and Inzirillo, 2024b). Recently, KAN-based convolutional neural networks have also been proposed which consist of kernels made of learnable non-linear functions that use splines (Bodner et al., 2024). These neural networks have been utilized for intrusion detection (Wang et al., 2025). KANs have also paved their way into celebrated architectures such as PointNet (Kashefi, 2024). Recent work has focused on improving different aspects of KANs such as making them more parameter-efficient (Ta et al., 2025), robust to overfitting (Altarabichi, 2024), and extending them to the complex domain (Wolff et al., 2025). More importantly, a framework was recently introduced which allows a seamless interplay between KANs and science (Liu et al., 2024). Specifically, multiplication was incorporated as a built-in modularity of KANs. Furthermore, the work strengthened the connection between KANs and scientific problems by highlighting the identification of key features, modular structures, and symbolic formulas.

Note that our study is not the only work in the intersection of KANs and graph learning algorithms (Kiamari et al., 2024; De Carlo et al., 2024; Zhang and Zhang, 2024; Li et al., 2024). However, while this paper provides an extensive evaluation across different settings and datasets, other works focus on a single task (e.g., node classification) or are limited to small-sized datasets. (Kiamari et al., 2024) propose two variants of the GCN architecture, where the feed-forward layer which updates node representations is replaced by KAN layers. The two variants differ from each other in how the new node representations are produced. The one transforms node representations before aggregation (and is thus similar to our `KAGCN` model), while the other after aggregation. However, the two variants are only evaluated on a single node classification dataset. (De Carlo et al., 2024) propose a neighborhood aggregation layer where spline-based activation functions are applied to the representations of the neighbors before aggregation, while another spline-based activation function is applied to the output of the aggregation. A final spline-based transformation then produces the final node representations. While the proposed model is evaluated in different tasks (i.e., node classification, graph classification and link prediction), all the considered datasets are small-sized. In another recent study, (Zhang and Zhang, 2024) also propose a variant of the GCN architecture, where a KAN is employed to transform node representations after aggregation. The proposed model is only evaluated on a single dataset where a graph is constructed from one-dimensional acoustic signals for axial flow pump fault diagnosis. Recent work has also investigated the ability of GNNs that employ KAN layers to solve problems in chemo- and bio-informatics such as the prediction of properties of molecules (Li et al., 2024) and the prediction of the binding affinity of small molecules to protein targets (Ahmed and Sifat, 2024), respectively. Such models were also applied to the problem of graph collaborative filtering (Xu et al., 2024b).

## 3    Background

### 3.1    Graph Learning Tasks

Before presenting the tasks on which we focus in this study, we start by introducing some key notation for graphs. Let $\mathbb{N}$ denote the set of natural numbers, i.e., $\{1, 2, \ldots\}$. Then, $[n] = \{1, \ldots, n\} \subset \mathbb{N}$ for $n \geq 1$.

Let $G = (V, E)$ be an undirected graph, where $V$ is the vertex set and $E$ is the edge set. We denote by $n$ the number of vertices and by $m$ the number of edges, i.e., $n = |V|$ and $m = |E|$. Let $g : V \to [n]$ denote a bijective mapping from the space of nodes to set $[n]$. Let $\mathcal{N}(v)$ denote the neighbourhood of vertex $v$, i.e., the set $\{u \mid \{v, u\} \in E\}$. The degree of a vertex $v$ is $\deg(v) = |\mathcal{N}(v)|$. Each node $v \in V$ is associated with a feature vector $\mathbf{x}_v \in \mathbb{R}^d$, and the feature matrix for all nodes is represented as $\mathbf{X} \in \mathbb{R}^{n \times d}$. Thus, $\mathbf{x}_v$ is equal to the $g(v)$-th row of $\mathbf{X}$.

In *node classification*, each node $v \in V$ is associated with a label $y_v$ that represents a class. The task is to learn how to map nodes to their class labels, i.e., to learn a function $f_{\text{node}}$ such that $f_{\text{node}}(v, G, \mathbf{X}) = y_v$. In *graph regression/classification*, the dataset consists of a collection of $N$ graphs $G_1, \ldots, G_N$ along with their class labels/targets $y_{G_1}, \ldots, y_{G_N}$. The task is then to learn a function that maps graphs to their class labels/targets, i.e., a function $f_{\text{graph}}$ such that $f_{\text{graph}}(G, \mathbf{X}) = y_G$, which can be discrete or continuous, for graph *classification* or graph *regression*, respectively.

The standard approach for learning such predictors (both for node- and graph-level tasks) is to first embed the nodes of the graph(s) into some vector space. That is, we aim to learn $\mathbf{H} = f_{\text{embedding}}(G, \mathbf{X}) \in \mathbb{R}^{n \times d_e}$ where $d_e$ denotes the embedding dimension. Then, the $g(v)$-th row of matrix $\mathbf{H}$ represents the embedding of node $v$. Let $\mathbf{h}_v$ denote this embedding. For node-level tasks, we can use $\mathbf{h}_v$ to predict directly the class label/target of node $v$. For graph-level tasks, we also need to apply a readout function on all the representations of the graph's nodes to obtain a representation $\mathbf{h}_G = f_{\text{readout}}(\mathbf{H})$ for the entire graph. One particularly desirable property of such models is permutation invariance. That is, the embedding $\mathbf{h}_G$ of a graph needs to be the same regardless of the ordering of its nodes. Indeed, these orderings do not hold any semantic meaning, and different orderings give rise to isomorphic graphs. Permutation invariance is achieved at the readout step by utilizing a permutation invariant operation over the rows of $\mathbf{H}$, such as the sum, max, or mean operators.

### 3.2 Graph Neural Networks

One of the most widely-used paradigms for designing such permutation invariant models is the message-passing framework (Gilmer et al., 2017) which consists of a sequence of layers and within each layer the embedding of each node is computed as a learnable function of its neighbors' embeddings. Formally, the embedding $\mathbf{h}_v^{(\ell)} \in \mathbb{R}^{d_\ell}$ at layer $\ell$ is computed as follows:

$$\mathbf{h}_v^{(\ell)} = \phi^{(\ell)} \left( \mathbf{h}_v^{(\ell-1)}, \bigoplus_{u \in \mathcal{N}(v)} \mathbf{h}_u^{(\ell-1)} \right) \tag{1}$$

where $\bigoplus$ is a permutation-invariant aggregation function (e.g., mean, sum), and $\phi^{(\ell)}$ is a differentiable function (e.g., linear transformation, MLP) that combines and transforms the node's previous embedding with the aggregated vector of its neighbors.

As discussed above, we focus here on the functions that different GNN models employ to update node representations. Some GNNs use a 1-layer perceptron (i.e., a linear mapping followed by a non-linear activation function) within each neighborhood aggregation layer to update node features (Duvenaud et al., 2015; Kipf and Welling, 2017; Zhang et al., 2018). For instance, each layer of the Graph Convolutional Network (GCN) (Kipf and Welling, 2017) is defined as follows:

$$\mathbf{h}_v^{(\ell)} = \sigma \left( \mathbf{W}^{(\ell)} \sum_{u \in \mathcal{N}(v) \cup \{v\}} \frac{\mathbf{h}_u^{(\ell-1)}}{\sqrt{(\deg(v) + 1)(\deg(u) + 1)}} \right) \tag{2}$$

where $\sigma$ is a non-linear activation and $\mathbf{W}^{(\ell)}$ is a trainable weight matrix.

However, the 1-layer perceptron is not a universal approximator of multiset functions (Xu et al., 2019), limiting the expressivity of the model. Thus, more recent models use MLPs instead of 1-layer perceptrons to update node representations (Balcilar et al., 2021; Dasoulas et al., 2021; Murphy et al., 2019; Nikolentzos et al., 2020). It is well-known that standard message-passing GNNs are bounded in expressiveness by the

Weisfeiler-Leman (WL) test of isomorphism (Xu et al., 2019). While two isomorphic graphs will always be mapped to the same representation by such a GNN, some non-isomorphic graphs might also be assigned identical representations.

A model that can achieve the same expressive power as the WL test, given sufficient width and depth of the MLP, is the Graph Isomorphism Network (GIN) (Xu et al., 2019), which is defined as follows:

$$\mathbf{h}_v^{(\ell)} = \text{MLP}^{(\ell)} \left( (1 + \epsilon^{(\ell)}) \cdot \mathbf{h}_v^{(\ell-1)} + \sum_{u \in \mathcal{N}(v)} \mathbf{h}_u^{(\ell-1)} \right) \tag{3}$$

where $\epsilon^{(\ell)}$ denotes a trainable parameter, and $\text{MLP}^{(\ell)}$ a trainable MLP.

The GIN model can achieve its full potential if proper weights (i.e., for the different $\text{MLP}^{(\ell)}$ layers and $\epsilon^{(\ell)}$) are learned, which is not guaranteed. This has motivated a series of works that focused on improving the training procedure of GNNs. For example, Ortho-GConv is an orthogonal feature transformation that can address GNNs' unstable training (Guo et al., 2022). Other works have studied how to initialize the weights of the MLPs of the message-passing layers of GNNs. It was shown that adopting the weights of converged MLPs as the weights of corresponding GNNs can lead to performance improvements in node classification tasks (Han et al., 2023). On the other hand, there exist settings where there is no need for complex learning models. This has led to the development of methods for simplifying GNNs. This can be achieved by removing the nonlinearities between the neighborhood aggregation layers and collapsing the resulting function into a single linear transformation (Wu et al., 2019) or by feeding the node features into a neural network which generates predictions and then propagating those predictions via a personalized PageRank scheme (Gasteiger et al., 2019).

Finally, the Graph Attention Network (GAT)(Veličković et al., 2017) leverages the multi-head attention mechanism among nodes to give each neighbor an adapted importance in the aggregation scheme.

$$\mathbf{h}_v^{(\ell)} = \sum_{u \in \mathcal{N}(v) \cup \{v\}} \alpha_{v,u}^{(\ell)} \mathbf{W}^{(\ell)} \mathbf{h}_u^{(\ell-1)} \tag{4}$$

where $\mathbf{W}^{(\ell)}$ denotes a single linear layer and

$$\alpha_{uv}^{(\ell)} = \frac{\exp\left( \text{LeakyReLU}\left( \left[ \mathbf{W}^{(\ell)} h_u^{(\ell-1)} + \mathbf{W}^{(\ell)} h_v^{(\ell-1)} \right] \right) \right)}{\sum_{u \in \mathcal{N}(v) \cup \{v\}} \left( \exp\left( \text{LeakyReLU}\left( \left[ \mathbf{W}^{(\ell)} h_u^{(\ell-1)} + \mathbf{W}^{(\ell)} h_v^{(\ell-1)} \right] \right) \right) \right)} \tag{5}$$

are the neighbor-wise attention weights for a given head.

### 3.3 Kolmogorov-Arnold Networks

### 3.3.1 Principle

Presented as an alternative to the MLP, the Kolmogorov-Arnold Network (KAN) architecture has recently attracted a lot of attention in the machine learning community (Liu et al., 2025). As mentioned above, this model relies on the Kolmogorov-Arnold representation theorem, which states that any multivariate function $f : [0, 1]^d \to \mathbb{R}$ can be written as:

$$f(\mathbf{x}) = \sum_{i=1}^{2d+1} \Phi_i \left( \sum_{j=1}^{d} \phi_{ij}(\mathbf{x}_j) \right) \tag{6}$$

where all $\Phi_\square$ and $\phi_\square$ denote univariate functions, and the sum is the only multivariate operator.

Equation 6 can be seen as a two-step process. First, a different set of univariate non-linear activation functions is applied to each dimension of the input, and then the output of those functions are summed up. The authors rely on this interpretation to define a Kolmogorov-Arnold Network (KAN) layer, which is a

mapping between a space $A \subseteq \mathbb{R}^d$ and a different space $B \subseteq \mathbb{R}^{d'}$, (identical in use to an MLP layer). Such a layer consists of $d \times d'$ trainable functions $\{\phi_{ij}, \ 1 \le i \le d', \ 1 \le j \le d\}$. Then, for $\mathbf{x} \in A$, we compute its image $\mathbf{x}'$ as:

$$\mathbf{x}'_i = \sum_{j=1}^{d} \phi_{ij}(\mathbf{x}_j) \tag{7}$$

Stacking two such layers, one with input dimension $d$ and output dimension $2d+1$, and another with input dimension $2d+1$ and output dimension 1, we obtain Equation 6, and the derived model is a universal function approximator. This seemingly offers a complexity advantage compared to MLPs, since the number of univariate functions required to represent any multivariate function from $[0,1]^d$ to $\mathbb{R}^{d'}$ is at most $(2d^2+d)\times d'$, whereas the universal approximation theorem for the MLP requires a possibly infinite number of neurons. However, as stated in the original paper, the behavior of such univariate functions might be arbitrarily complex (e.g., fractal, non-smooth), thus leading to them being non-representable and non-learnable.

MLPs relax the infinite-width constraint by stacking finite-width layers. Likewise, KANs relax the arbitrary complexity constraints on the non-linearities by stacking KAN layers of lower complexity. Thus, the output of a function is given by:

$$y = \mathtt{KAN}(\mathbf{x}) = \Phi_L \circ \Phi_{L-1} \circ \cdots \circ \Phi_1(\mathbf{x}) \quad \text{where } \Phi_1, \ldots, \Phi_L \text{ are KAN layers.} \tag{8}$$

### 3.3.2 KAN Implementations and Applications

The original paper that introduced KANs uses splines (i.e., trainable piecewise-polynomial functions) as nonlinearities. This allows to retain a high expressivity for a relatively small number of parameters, at the cost of enforcing some local smoothness. A layer $\ell$ is thus a $d_\ell \times d_{\ell-1}$ grid of splines. The degree used for each spline (called *spline order*), as well as the number of splines used for each function (called *grid size*) are both hyperparameters of the architecture. Nonetheless, KANs allow for any function to serve as its base element. Thus, alternative implementations use different bases, such as radial-basis functions (Li, 2024), which in particular alleviate some computational bottleneck of splines.

## 4 KAN-based GNN Layers

We next derive three $\mathtt{KAGNN}$ layers, i.e. variants of the GIN, GCN and GAT models which use KANs to transform the node features instead of fully-connected layers or MLPs.

### 4.1 The KAGIN Layer

To achieve its maximal expressivity, the GIN model relies on the MLP architecture and its universal approximator property. Since KAN is also a universal function approximator, we could achieve the same expressive power using KANs in lieu of MLPs. We thus propose the $\mathtt{KAGIN}$ model, defined as:

$$\mathbf{h}_v^{(\ell)} = \mathtt{KAN}^{(l)} \left( (1 + \epsilon) \cdot \mathbf{h}_v^{(\ell-1)} + \sum_{u \in \mathcal{N}(v)} \mathbf{h}_u^{(\ell-1)} \right) \tag{9}$$

With theoretically-sound KANs (i.e., with arbitrarily complex components), this architecture is exactly as expressive as the vanilla GIN model with infinite layer width. In practice, we employ two different families of functions to serve as the base components, leading to two different layers: $\mathtt{BS\text{-}KAGIN}$, based on B-splines, as in the original paper; and $\mathtt{RBF\text{-}KAGIN}$, based on radial basis functions (RBFs), which were proposed as a more computationally-efficient alternative. While these do not guarantee universal approximation, the empirical results in the original paper demonstrate the great expressive power of KANs (Liu et al., 2025), motivating our empirical study.

### 4.2 The `KAGCN` Layer

GCN-based architectures have achieved great success in node classification tasks. While in our experiments we evaluate `KAGIN` on node classification datasets, the objective advantage of GCN over GIN on some of the datasets does not facilitate a fair estimation of KANs' potential in this context. To this end, we also propose a variant of the GCN model. Specifically, we substitute the linear transformation of the standard GCN (Kipf and Welling, 2017) model with a single KAN layer (defined in Equation 8) to obtain the `KAGCN` layer:

$$\mathbf{h}_v^{(\ell)} = \Phi^{(\ell)} \left( \sum_{u \in \mathcal{N}(v) \cup \{v\}} \frac{\mathbf{h}_u^{(\ell-1)}}{\sqrt{(\deg(v)+1)(\deg(u)+1)}} \right) \tag{10}$$

where $\Phi^{(\ell)}$ denotes a single KAN layer. In the familiar matrix formulation, where $\tilde{\mathbf{A}} = \mathbf{A} + \mathbf{I}$ is the adjacency matrix with self-loops and $\tilde{\mathbf{D}}$ the diagonal degree matrix of $\tilde{\mathbf{A}}$, the node update rule of `KAGCN` can be written as $\mathbf{H}^{(\ell)} = \Phi^{(\ell)} \left( \tilde{\mathbf{D}}^{-\frac{1}{2}} \tilde{\mathbf{A}} \tilde{\mathbf{D}}^{-\frac{1}{2}} \mathbf{H}^{(\ell-1)} \right)$ where the different rows of $\mathbf{H}^{(\ell)}$ store the representations of the different nodes of the graph. We propose both a `BS-KAGCN` and a `RBF-KAGCN`, where $\Phi$ is based respectively on B-splines and RBFs.

### 4.3 The `KAGAT` Layer

Finally, we substitute the linear transformation of the standard graph attention network (GAT) (Veličković et al., 2017) model with a single KAN layer (defined in Equation 8) to obtain the `KAGAT` layer:

$$\mathbf{h}_v^{(\ell)} = \sum_{u \in \mathcal{N}(v) \cup \{v\}} \alpha_{v,u}^{(\ell)} \Phi^{(\ell)} \left( \mathbf{h}_u^{(\ell-1)} \right) \tag{11}$$

where $\Phi^{(\ell)}$ denotes a single KAN layer and

$$\alpha_{uv}^{(\ell)} = \frac{\exp \left( \texttt{LeakyReLU} \left( \left[ \Phi^{(\ell)} \left( h_u^{(\ell-1)} \right) + \Phi^{(\ell)} \left( h_v^{(\ell-1)} \right) \right] \right) \right)}{\sum_{u \in \mathcal{N}(v) \cup \{v\}} \left( \exp \left( \texttt{LeakyReLU} \left( \left[ \Phi^{(\ell)} \left( h_u^{(\ell-1)} \right) + \Phi^{(\ell)} \left( h_v^{(\ell-1)} \right) \right] \right) \right) \right)} \tag{12}$$

## 5 Empirical Evaluation

In this section, we compare all three `KAGNN` models against MLP-based GNNs on the following tasks: node classification, link prediction, graph classification, and graph regression. All models are implemented with PyTorch (Paszke et al., 2019). For KAN layers, we rely on publicly available implementations for B-splines [1] and for RBF [2]. For a given task, all models have consistent architectures, differing as little as possible. All GAT-based models use 4 heads.

### 5.1 Node classification

**Datasets** To evaluate the performance of GNNs with KAN layers in the context of node classification, we use 7 well-known datasets of varying sizes and types, including homophilic (Cora, Citeseer (Kipf and Welling, 2017) and Ogbn-arxiv (Hu et al., 2020)) and heterophilic (Cornell, Texas, Wisconsin, Actor) networks (Zhu et al., 2021). The homophilic networks are already split into training, validation, and test sets, while the heterophilic datasets are accompanied by fixed 10-fold cross-validation indices. Thus, we clone the split for homophilic datasets 10 times in order to have a similar number of experiments across all datasets.

**Experimental setup.** We perform two types of experiments. First, we compare our KAGCN model on the homophilic datasets to the GCN proposed in the state of the art approach in (Luo et al., 2024). The

---

[1] https://github.com/Blealtan/efficient-kan
[2] https://github.com/ZiyaoLi/fast-kan

Table 1: Average classification accuracy (± standard deviation) and model parameters based on Luo et al.(Luo et al., 2024) using GCN-based models on the Cora, CiteSeer, and Arxiv datasets.

| Model | Cora | | CiteSeer | | Arxiv | |
|---|---|---|---|---|---|---|
| | Accuracy (%) | Params | Accuracy (%) | Params | Accuracy (%) | Params |
| GCN | $81.60 \pm 0.40$ | 3,262,983 | $\mathbf{71.60} \pm \mathbf{0.40}$ | 6,221,830 | $\mathbf{71.74} \pm \mathbf{0.29}$ | 2,591,784 |
| BS-KAGCN | $80.84 \pm 1.61$ | 3,702,144 | $70.04 \pm 1.66$ | 4,317,952 | $70.58 \pm 0.20$ | 1,010,432 |
| RBF-KAGCN | $\mathbf{81.72} \pm \mathbf{0.79}$ | 3,714,217 | $68.16 \pm 0.61$ | 6,258,808 | $71.39 \pm 0.28$ | 813,647 |

Table 2: Average node classification accuracies (± standard deviation).

| Model | Cora | Citeseer | Ogbn-arxiv | Cornell | Texas | Wisconsin | Actor |
|---|---|---|---|---|---|---|---|
| GCN | $74.01 \pm 0.81$ | $57.73 \pm 1.21$ | $53.21 \pm 0.13$ | $66.13 \pm 6.09$ | $72.34 \pm 5.92$ | $78.63 \pm 5.43$ | $32.81 \pm 1.11$ |
| BS-KAGCN | $70.46 \pm 2.00$ | $56.45 \pm 1.62$ | $\mathbf{53.76} \pm 0.31$ | $67.75 \pm 6.54$ | $72.16 \pm 5.68$ | $78.63 \pm 5.26$ | $34.45 \pm 1.12$ |
| RBF-KAGCN | $48.06 \pm 1.40$ | $47.55 \pm 1.43$ | $53.65 \pm 0.20$ | $67.75 \pm 7.24$ | $71.62 \pm 4.69$ | $77.45 \pm 4.11$ | $\mathbf{34.77} \pm 1.20$ |
| GAT | $76.51 \pm 0.79$ | $\mathbf{60.91} \pm 1.04$ | $53.68 \pm 0.12$ | $66.85 \pm 5.59$ | $69.01 \pm 6.14$ | $78.17 \pm 5.70$ | $32.51 \pm 1.17$ |
| BS-KAGAT | $\mathbf{78.28} \pm 0.83$ | $56.87 \pm 1.67$ | $55.17 \pm 0.12$ | $67.66 \pm 6.94$ | $\mathbf{73.78} \pm 6.39$ | $78.43 \pm 5.47$ | $31.66 \pm 1.23$ |
| RBF-KAGAT | $72.62 \pm 1.64$ | $48.65 \pm 2.39$ | $54.24 \pm 0.25$ | $64.14 \pm 6.14$ | $71.71 \pm 6.49$ | $76.34 \pm 4.34$ | $34.04 \pm 0.94$ |
| GIN | $70.85 \pm 1.75$ | $47.98 \pm 0.91$ | $53.58 \pm 0.74$ | $69.10 \pm 4.85$ | $73.51 \pm 5.87$ | $78.24 \pm 4.76$ | $33.46 \pm 1.38$ |
| BS-KAGIN | $65.78 \pm 1.68$ | $55.77 \pm 1.51$ | $53.05 \pm 0.55$ | $\mathbf{69.64} \pm 7.40$ | $72.97 \pm 5.63$ | $\mathbf{78.63} \pm 5.16$ | $34.26 \pm 1.06$ |
| RBF-KAGIN | $72.77 \pm 1.35$ | $46.87 \pm 1.93$ | $53.56 \pm 0.37$ | $67.30 \pm 6.44$ | $73.60 \pm 5.05$ | $76.73 \pm 4.88$ | $34.37 \pm 0.95$ |

results are shown in Table 1. One can observe that the classical GCN outperforms the KAN-based model in 2 out of 3 datasets, albeit with a narrow difference and confidence overlaps in all of the comparisons. Note that the current setting is built to favor the classical GNNs, as it relies on techniques such as dropout, jumping knowledge, batch normalization and preliminary linear transformations. Hence, although the MLP-based GCN enjoys a plethora of assistive techniques examined in the past decades, the KAN-based counterpart exhibits a close performance. We should note here that these assistive techniques have not been developed yet for KAN-based architectures, and our results aim at motivating the development of such techniques.

Our second approach compares the vanilla versions of the neural architectures when we replace the MLP with KAN, as delineated in sections 4. In order to clarify the effect of KAN on the model's performance, we only use dropout and batchnorm as additional techniques. Indeed, we aspire to compare the classical MLP-based GNNs as if they were in the same nascent stage as their KAN-based counterparts.

In order for all models to exploit the same quantity of information, we fix the number of message-passing layer dataset-wise, with 2 for Cora and CiteSeer, 3 for Texas, Cornell, Wisconsin, and Ogbn-arxiv and 4 for Actor. All attention-based models used 4 heads.

For every dataset and model, we tune the values of the hyperparameters using the Optuna package (Akiba et al., 2019) with 100 trials (parameterizations), a TPE Sampler and set early stopping patience to 50. For each parameterization, and for each split, we train 3 models in order to mitigate randomness. We then select the parameterization with lowest validation error across all splits and models.

We use a different hyperparameter range for MLP-based models, B-Splines-based models and RBF-based models (see Table 8). Indeed, for the same number of "neurons", a KAN-based layer will have a much larger number of parameters than an MLP-based layer (see Appendix C.1 for examples). In order to avoid giving an unfair advantage to KAN-based layers, we thus allowed the MLP layers to be larger.

**Results.** The results are given in Table 2. We see that a B-splines-based variant outperforms other models on 5/7 datasets, and both other architecture win on one dataset each. Nonetheless, these are often by small margins. We also notice poor performance of RBF-KAGCN on Cora, which we attribute to overfitting. Overall,there seems to be a positive impact of using KAN-based architectures.

Table 3: AUC for link prediction of the `KAGIN` and GIN models on the link-prediction datasets.

| | Cora | CiteSeer | | Cora | CiteSeer | | Cora | CiteSeer |
|---|---|---|---|---|---|---|---|---|
| GCN | **95.36** $\pm$ 0.23 | 96.72 $\pm$ 0.44 | GAT | 94.02 $\pm$ 0.48 | 96.10 $\pm$ 0.49 | GIN | 94.97 $\pm$ 0.40 | 95.37 $\pm$ 0.23 |
| BS-KAGCN | 94.37 $\pm$ 0.46 | 94.91 $\pm$ 0.62 | BS-KAGAT | 94.71 $\pm$ 0.65 | 95.40 $\pm$ 0.33 | BS-KAGIN | 94.64 $\pm$ 0.44 | 95.75 $\pm$ 0.39 |
| RBF-KAGCN | 94.87 $\pm$ 0.45 | 95.44 $\pm$ 0.67 | RBF-KAGAT | 94.96 $\pm$ 0.42 | 90.09 $\pm$ 14.1 | RBF-KAGIN | 95.07 $\pm$ 0.27 | **97.07** $\pm$ 0.40 |

## 5.2 Link Prediction

**Datasets and Experimental Setup.**  In this setting, we focus on the task of link prediction. We use two datasets, Cora and CiteSeer, following the implementation and protocol provided in (Li et al., 2023; Mao et al., 2024). We performed a grid-search over learning rate, hidden dimension, number of hidden layers, grid size and spline order, each time training models on 10 splits. We then selected the model with lowest validation loss to obtain the best configuration, and reported its test ROC-AUC (over all 10 splits).

**Results.**  Table 3 displays the results on the link prediction task. We see that GCN is the best performer on Cora, and `RBF-KAGIN` is the best on CiteSeer, albeit with often close scores among models. We also notice some instability of `RBF-KAGAT` on CiteSeer.

## 5.3 Graph Classification

**Datasets.**  In this set of experiments, we evaluate the `KAGNN` models on standard graph classification benchmark datasets (Morris et al., 2020b). We experiment with the following 7 datasets: MUTAG, DD, NCI1, PROTEINS, ENZYMES, IMDB-B, IMDB-M.

**Experimental setup.** We follow the experimental protocol proposed in (Errica et al., 2020). We kept our architecture simple: first, a number of message-passing layers extract node representation, whose model is either an MLP (resp. a KAN). We then use pooling (sum-pooling for GINs and GATs, mean-pooling for GCNs) to obtain graph representations. Finally, an MLP (resp. a KAN) with same architecture as those in the message-passing layer, is used as a classifier. A softmax is used for predicting class probabilities. Thus, we perform 10-fold cross-validation, while within each fold a model is selected based on a 90%/10% split of the training set. We use the pre-defined splits provided in (Errica et al., 2020). We use the Optuna package to select the model that achieves the lowest validation cross-entropy loss over 100 iterations. For fair comparison, we fix the number of message-passing layers of all architectures for each dataset. On MUTAG, PROTEINS, IMDB-B and IMDB-M, we set the number of layers to 2. On DD, we set it to 3. On ENZYMES, we set it to 4 and finally, on NCI1, we set it to 5. We train each model for $1,000$ epochs (early stopping with a patience of 20 epochs) by minimizing the cross entropy loss. We use the Adam optimizer for model training (Kingma and Ba, 2015). We apply batch normalization (Ioffe and Szegedy, 2015) and dropout (Srivastava et al., 2014) to the output of each message-passing layer. The ranges for hyperparameters search are given in table 11. Once the best hyperparameters are found for a given split, we train 3 different models on the training set of the split and evaluate them on the test set of the split. This yields 3 test accuracies for this split, and we compute their average to obtain the performance for this split.

**Results.**  Table 4 illustrates the average classification accuracies and the corresponding standard deviations of the three models on the different datasets. We observe that B-splines variants win on 4 datasets, and the MLP-based GIN leads on 3, albeit by small margins wrt. the standard deviation. We notice that, while the MLP-based GCN performs poorly (predictably so, since graph-level tasks often require higher complexity than node-classification ones), the KAN-based GCN perform significantly better. This tends to confirm that a single KAN layer has much more representation power than a fully-connected layer.

## 5.4 Graph Regression

**Datasets.**  We experiment with two molecular datasets: (1) ZINC-12K (Irwin and Shoichet, 2005), and (2) QM9 (Ramakrishnan et al., 2014). ZINC-12K consists of $12,000$ molecules. The task is to predict the

Table 4: Average graph classification accuracy ($\pm$ standard deviation).

| | MUTAG | DD | NCI1 | PROTEINS | ENZYMES | IMDB-B | IMDB-M |
|---|---|---|---|---|---|---|---|
| GCN | $59.95 \pm 8.98$ | $65.87 \pm 6.33$ | $60.10 \pm 4.63$ | $63.10 \pm 3.75$ | $20.56 \pm 3.70$ | $68.40 \pm 4.01$ | $48.56 \pm 3.71$ |
| BS-KAGCN | $71.70 \pm 7.54$ | $72.74 \pm 4.16$ | $73.97 \pm 4.01$ | $\mathbf{75.68} \pm 3.49$ | $\mathbf{59.06} \pm 4.48$ | $72.87 \pm 5.12$ | $50.04 \pm 3.99$ |
| RBF-KAGCN | $77.86 \pm 9.45$ | $66.78 \pm 5.91$ | $73.07 \pm 2.90$ | $73.67 \pm 3.99$ | $56.83 \pm 8.37$ | $73.13 \pm 4.5$ | $49.84 \pm 3.78$ |
| GAT | $57.51 \pm 12.3$ | $71.92 \pm 5.88$ | $60.42 \pm 8.21$ | $59.78 \pm 0.87$ | $18.17 \pm 1.34$ | $67.37 \pm 3.83$ | $48.24 \pm 5.02$ |
| BS-KAGAT | $82.66 \pm 4.78$ | $\mathbf{78.27} \pm 2.37$ | $76.46 \pm 3.66$ | $72.75 \pm 2.18$ | $51.33 \pm 12.9$ | $72.00 \pm 3.80$ | $50.09 \pm 4.34$ |
| RBF-KAGAT | $75.88 \pm 12.1$ | $73.42 \pm 4.76$ | $72.63 \pm 4.65$ | $71.93 \pm 4.23$ | $49.78 \pm 7.65$ | $71.00 \pm 5.04$ | $48.13 \pm 4.15$ |
| GIN | $\mathbf{86.16} \pm 5.96$ | $74.78 \pm 3.82$ | $79.32 \pm 1.63$ | $70.66 \pm 3.60$ | $54.72 \pm 6.55$ | $\mathbf{73.37} \pm 2.90$ | $\mathbf{50.36} \pm 4.10$ |
| BS-KAGIN | $85.79 \pm 8.06$ | $76.06 \pm 5.10$ | $\mathbf{80.34} \pm 1.46$ | $73.37 \pm 2.81$ | $44.50 \pm 8.53$ | $73.13 \pm 4.46$ | $50.33 \pm 3.87$ |
| RBF-KAGIN | $85.10 \pm 5.60$ | $75.92 \pm 3.49$ | $79.19 \pm 1.81$ | $74.33 \pm 3.52$ | $44.78 \pm 9.58$ | $69.93 \pm 3.84$ | $49.22 \pm 4.41$ |

Table 5: Average MAE ($\pm$ standard deviation) of the KAGIN and GIN models on graph regression.

| | ZINC-12K | QM9 |
|---|---|---|
| GIN | $0.4131 \pm 0.0215$ | $0.0969 \pm 0.0017$ |
| BS-KAGIN | $\mathbf{0.3000} \pm 0.0332$ | $\mathbf{0.0618} \pm 0.0007$ |
| RBF-KAGIN | $0.3026 \pm 0.0663$ | $0.0778 \pm 0.0023$ |

constrained solubility of molecules, an important property for designing generative GNNs for molecules. The dataset is already split into training, validation and test sets ($10,000$, $1,000$ and $1,000$ graphs in the training, validation and test sets, respectively). QM9 contains approximately $134,000$ organic molecules. Each molecule consists of Hydrogen, Carbon, Oxygen, Nitrogen, and Flourine atoms and contain up to 9 heavy (non Hydrogen) atoms. The task is to predict 12 target properties for each molecule. The dataset was divided into a training, a validation, and a test set according to a $80\%/10\%/10\%$ split.

**Experimental setup.** We perform grid search to select values for the different hyperparameters. We choose the number of hidden layers from $\{2, 3, 4\}$ for GIN and $\{1, 2, 3\}$ for KAN. We use different ranges for each model was because we wanted to limit the size advantage of KAN against MLP, since a layer with the same width of KAN has more parameters than an MLP one). For both models we search a learning rate from $\{10^{-3}, 10^{-4}\}$. For GIN, we choose the hidden dimension size from $\{32, 64, 128, 256, 512, 1024\}$, while for KAGIN, we choose it from $\{4, 8, 16, 32, 64, 128, 256\}$. For KAGIN, we also select the grid size from $\{1, 3, 5, 8, 10\}$ and for BS-KAGIN the spline order from $\{3, 5\}$. To produce graph representations, we use the sum operator. The emerging graph representations are finally fed to a 2-layer MLP (for GIN) or a 2-layer KAN (for KAGIN) which produces the output. We set the batch size equal to 128 for all models. We train each model for $1,000$ epochs by minimizing the mean absolute error (MAE). We use the Adam optimizer for model training (Kingma and Ba, 2015). We also use early stopping with a patience of 20 epochs. For ZINC-12K, we also use an embedding layer that maps node features into 100-dimensional vectors. We choose the configuration that achieves the lowest validation error. Once the best configuration is found, we run 10 experiments and report the average performance on the test set. For both datasets and models, we set the number of message-passing layers to 4. On QM9, we performed a joint regression of the 12 targets.

**Results.** The results are shown in Table 5. We observe that on both considered datasets, both KAGIN variants significantly outperform the GIN model. Note that these datasets are significantly larger (in terms of the number of samples) compared to the graph classification datasets of Table 4. The results also indicate that BS-KAGIN outperforms RBF-KAGIN on both ZINC-12K and QM9. However, the difference in performance between the two models is very small. Thus, it appears that both B-splines and RBFs are good candidates as KANs' base components in regression tasks. In addition, we should note that the BS-KAGIN model offers an absolute improvement of approximately 0.11 and 0.03 in MAE over GIN. Those improvements suggest that KANs might be more effective than MLPs in regression tasks.

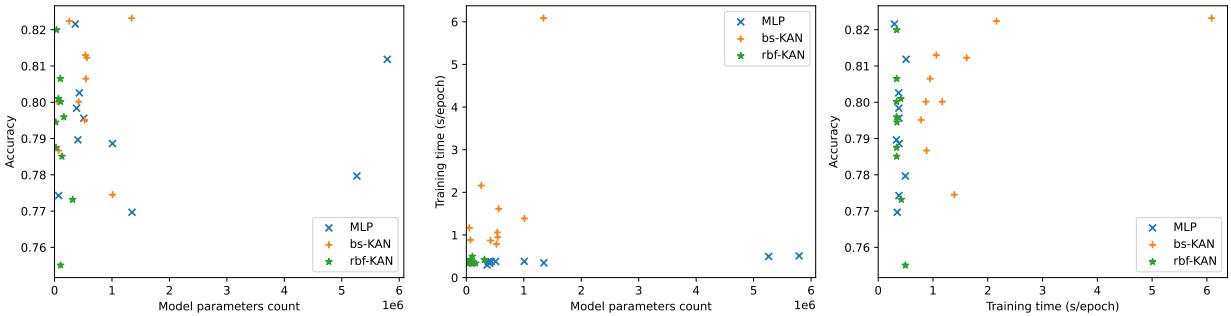

Figure 1: Parameters count/size/performance comparison, GIN, NCI1

# 6 Analysis of Size and Time

## 6.1 Setting

Following the above results, and it is legitimate to wonder whether the performance of the KAN models comes from a larger size than their MLP-based counterparts. We show in this section that it is not the case. Using the graph-classification models trained for the results provided in 5.3, we display here three relations: the size-to-performance relation, the size-to-training time relation and the training time-to-performance relation. Note that these models are the ones selected by the hyperparameter search algorithm, thus exhibiting the best validation performance throughout the architecture-optimization process.

We thus notice an interesting tradeoff, where RBF-based KANs tend to train models with good performance, but smaller size than their MLP-based counterparts, allowing them to train almost as quickly. Meanwhile BS-based KANs tend to be slower.

## 6.2 Discussion

Figures 1 shows these relations on dataset NCI1 for the GIN-based models, a setting where the three models have a similar performance. We also show in table 6 the associated numerical values for this dataset. The rest of the graph classification datasets and models are provided in App. C.1. We notice that, for the same number of parameters, MLPs are the fastest, followed closely by RBF-based KANs; then BS-based KANs are significantly slower than both others. This confirms the fact that the RBF-based representation alleviates greatly the computational bottlenecks of B-splines. Nonetheless, we also notice that, despite the hyperparameter search setting allowing them a large size, the performance-driven model selection converged towards smaller sizes for KAN-based models than for MLPs. This allows the RBF-based version of `KAGNN` models to train in time similar to their MLP-based counterpart. Nonetheless, the BS-based variant remains significantly slower in almost all settings.

# 7 Discussion and Perspectives

**Prominent application domains.** KANs, both in our work and in the recent literature, seem to be a promising alternative to MLPs for regression tasks. They rely on combinations of continuous functions which could explain their improved performance for continuous outputs. Moreover, this property and their empirical performance in regression tasks renders them an ideal candidate for physics-inspired architectures (Karniadakis et al., 2021), which address problems of high importance, from climate forecasting (Kashinath et al., 2021) to atomic simulations (Batzner et al., 2022) and rely heavily on graph-based architectures.

**Explainability matters.** Recent works on KANs argue that their potential for explainability is higher than that of MLPs with post-hoc methods (e.g. Shapley values, gradient-based) (Liu et al., 2025), since closed-form symbolic formulas can be drawn from the former (Liu et al., 2024), providing complete overview

Table 6: Average time and parameter sizes for selected models, NCI1

| Model | Parameter count | Accuracy | time per epoch (s) |
|---|---|---|---|
| GCN | 316588.40 | $0.60 \pm 0.04$ | 0.31 |
| BS-KAGCN | 182464.80 | $0.74 \pm 0.04$ | 0.94 |
| RBF-KAGCN | 117043.10 | $0.73 \pm 0.03$ | 0.37 |
| GAT | 1566334.00 | $0.60 \pm 0.08$ | 0.47 |
| BS-KAGAT | 1109714.80 | $0.76 \pm 0.04$ | 1.42 |
| RBF-KAGAT | 1468335.80 | $0.73 \pm 0.04$ | 0.53 |
| GIN | 1557454.00 | $0.79 \pm 0.02$ | 0.39 |
| BS-KAGIN | 532684.50 | $0.80 \pm 0.01$ | 1.70 |
| RBF-KAGIN | 107075.40 | $0.79 \pm 0.02$ | 0.37 |

of the underlying computations. Hence, KAN-based models could prove more friendly to practitioners from application fields, such as biology and physics, who tend to be skeptical of the black-box approach.

**GNNs for fixed graphs inspired from KANs.** Each KAN architecture consists of a set of neurons. Each one of those neurons is connected to other neurons. Thus, KANs can be naturally modeled as graphs where nodes represent neurons and edges represent the connections between them. Each layer of a KAN is actually a complete bipartite graph, while a KAN has learnable activation functions on the edges of its associated graph. There actually exist datasets that contain multiple graphs which share the same graph structure, but differ in the node and edge features. Two examples of such datasets are the MNIST and CIFAR10 graph classification datasets (Dwivedi et al., 2023). Since the graph structure is fixed (and identical for all graphs), we could associate each edge of the graph with one or more learnable activation functions (shared across graphs), similar to what KANs do. This would allow the model to learn different functions for each edge of the graph, leading potentially to high-quality node and graph representations.

**Advantages and future work.** Finally, we discuss potential advantages that KANs might have over MLPs, and leave their investigation for future work. First, their ability to accurately fit smooth functions could prove relevant on datasets where variables interact with some regular patterns. Second, their interpretability could be leveraged to provide explanations on learned models, giving insights into the nature of interactions among entities. Another study direction could be to extend the KAN architecture to more GNN models (e.g. GAT, GraphSAGE, Graph Transformer) to develop a wider KAN-based arsenal for graph tasks. Finally, a thorough study of the effect of the different hyperparameters could be pursued, allowing to fully exploit the splines or radial basis functions while retaining small networks.

## 8 Conclusion

In this paper, we have investigated the potential of Kolmogorov-Arnold networks in graph learning tasks. Since the KAN architecture is a natural alternative to the MLP, we developed three GNN architectures, `KAGCN`, `KAGAT` and `KAGIN`, respectively analogous to the GCN, GAT and GIN models. We then compared those architectures against each other in both node- and graph-level tasks. For link prediction, as well as graph and node classification, there does not appear to be a clear winner. For graph regression, our results indicate that KAN has an advantage over MLP, with the B-spline variant proving superior in terms of performance. Nonetheless, this gain comes at the cost of computational efficiency. In this regard, the RBF-based variant offers a lesser performance improvement, but proves much more efficient in terms of training time.

This paper shows that such KAN-based GNNs are valid alternatives to the traditional MLP-based models. We thus believe that these models deserve the attention of the graph machine-learning community, and that future work should be carried out to identify their specific advantages and weaknesses, allowing for an enlightened choice between KANs and MLPs based on the task and context at hand.

## Acknowledgements

This work was partially supported by the Wallenberg AI, Autonomous Systems and Software Program (WASP) funded by the Knut and Alice Wallenberg Foundation and partner Swedish universities and industry. Part of the experiments was enabled by resources provided by the National Academic Infrastructure for Supercomputing in Sweden (NAISS), partially funded by the Swedish Research Council through grant agreement no. 2022-06725. Part of the experiments was also enabled by the Berzelius resource, which was made possible through application support provided by National Supercomputer Centre at Linköping University.

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

# A    Further Graph Classification Experiments

Besides the graph classification experiments that are presented in the main paper, we also experimented with the Peptides-func dataset, a graph classification dataset from the Long Range Graph Benchmark (Dwivedi et al., 2022). The dataset consists of $15,535$ graphs where the average number of nodes per graph is equal to $150.94$. We evaluated GIN, BS-KAGIN and RBF-KAGIN on this dataset. We performed a grid search where for all models we chose the learning rate from $\{0.001, 0.0001\}$, the number of hidden layers of MLPs and KANs from $\{1, 2, 3\}$ and the hidden dimension from $\{64, 128\}$. We set the number of neighborhood aggregation layers to 5 for all models. For both BS-KAGIN and RBF-KAGIN, grid size was chosen from $\{3, 5\}$, and for BS-KAGIN, spline order was chosen from $\{3, 5\}$. Each experiment was run 4 times with 4 different seeds and Table 7 illustrates the average precision achieved by the three models.

Table 7: Average Precision ($\pm$ standard deviation) of the KAGIN and GIN models on Peptides-func

|  | Peptides-func |
| --- | --- |
| GIN | $55.01 \pm 0.75$ |
| BS-KAGIN | $\mathbf{58.71} \pm 1.02$ |
| RBF-KAGIN | $57.68 \pm 2.05$ |

We observe that BS-KAGIN and RBF-KAGIN outperform GIN on the Peptides-func dataset. Both models offer an absolute improvement of more than $2.5\%$ in average precision over GIN. Thus, even in tasks where long-range interactions between nodes need to be captured, for a fixed receptive field, KAN layers can help the models better capture those interactions.

## B   Hyperparameter search ranges

Table 8: Hyperparameter ranges for node classification (datasets Cora, Cornell, Texas, Wisconsin)

| Model | Learning Rate | Hidden Layers | Hidden Dim. | Grid Size | Spl. Order | Dropout |
|---|---|---|---|---|---|---|
| GIN | $[1e-5, 1e-2]$ | $[1, 4]$ | $[1, 256]$ | NA | NA | $[0, 0.9]$ |
| BS-KAGIN/KAGCN | $[1e-5, 1e-2]$ | $[1, 4]$ | $[1, 128]$ | $[1, 8]$ | $[1, 3]$ | $[0, 0.9]$ |
| RBF-KAGIN/KAGCN | $[1e-5, 1e-2]$ | $[1, 4]$ | $[2, 128]$ | $[2, 32]$ | NA | $[0, 0.9]$ |

Table 9: Hyperparameter ranges for node classification (datasets CiteSeer, Ogbn-Arxiv, Actor)

| Model | Learning Rate | Hidden Layers | Hidden Dim. | Grid Size | Spl. Order | Dropout |
|---|---|---|---|---|---|---|
| GIN | $[1e-5, 1e-2]$ | $[1, 4]$ | $[1, 64]$ | NA | NA | $[0, 0.9]$ |
| BS-KAGIN/KAGCN | $[1e-5, 1e-2]$ | $[1, 4]$ | $[2, 32]$ | $[1, 8]$ | $[1, 3]$ | $[0, 0.9]$ |
| RBF-KAGIN/KAGCN | $[1e-5, 1e-2]$ | $[1, 4]$ | $[2, 32]$ | $[2, 4]$ | NA | $[0, 0.9]$ |

Table 10: Hyperparameter ranges for link prediction

| Model | Learning Rate | Hidden Layers | Hidden Dim. | Grid Size | Spl. Order |
|---|---|---|---|---|---|
| GIN | $\{1e-2, 1e-3, 1e-4\}$ | $\{2, 4\}$ | $\{16, 512\}$ | NA | NA |
| BS-KAGIN/KAGCN | $\{1e-2, 1e-3\}$ | $\{2, 4\}$ | $\{16, 64\}$ | $\{8\}$ | $[2, 3\}]$ |
| RBF-KAGIN/KAGCN | $\{1e-2, 1e-3\}$ | $\{2, 4\}$ | $\{16, 512\}$ | $\{2, 16\}$ | NA |

Table 11: Hyperparameter ranges for graph classification

| Model | Learning Rate | Hidden Layers | Hidden Dim. | Dropout | Grid Size | Spl. Order |
|---|---|---|---|---|---|---|
| GIN | $[1e-4, 1e-2]$ | $[1, 4]$ | $[2, 512]$ | $[0, 0.9]$ | NA | NA |
| BS-KAGIN/KAGCN | $[1e-4, 1e-2]$ | $[1, 4]$ | $[2, 64]$ | $[0, 0.9]$ | $[2, 32]$ | $[1, 4]$ |
| RBF-KAGIN/KAGCN | $[1e-4, 1e-2]$ | $[1, 4]$ | $[2, 64]$ | $[0, 0.9]$ | $[2, 16]$ | NA |

## C  Time/Performance/Accuracy analysis

Figures 2 to 22 and tables 12 to 17 regarding the time/size/accuracy analysis on the graph classification task.

Table 12: Average time and parameter sizes for selected models, MUTAG

| Model | Parameter count | Accuracy | time per epoch (s) |
|---|---|---|---|
| GCN | 66421.40 | $0.60 \pm 0.09$ | 0.02 |
| BS-KAGCN | 31991.40 | $0.72 \pm 0.07$ | 0.03 |
| RBF-KAGCN | 67221.20 | $0.78 \pm 0.09$ | 0.02 |
| GAT | 1162504.80 | $0.58 \pm 0.12$ | 0.02 |
| BS-KAGAT | 521626.00 | $0.83 \pm 0.05$ | 0.04 |
| RBF-KAGAT | 1119860.90 | $0.76 \pm 0.11$ | 0.02 |
| GIN | 840858.00 | $0.86 \pm 0.06$ | 0.02 |
| BS-KAGIN | 145455.00 | $0.86 \pm 0.08$ | 0.05 |
| RBF-KAGIN | 82645.40 | $0.85 \pm 0.05$ | 0.02 |

Table 13: Average time and parameter sizes for selected models, DD

| Model | Parameter count | Accuracy | time per epoch (s) |
|---|---|---|---|
| GCN | 274320.80 | $0.66 \pm 0.06$ | 0.24 |
| BS-KAGCN | 110847.70 | $0.73 \pm 0.04$ | 0.93 |
| RBF-KAGCN | 110976.90 | $0.67 \pm 0.06$ | 0.24 |
| GAT | 224323.60 | $0.72 \pm 0.06$ | 0.22 |
| BS-KAGAT | 1375974.80 | $0.78 \pm 0.02$ | 2.36 |
| RBF-KAGAT | 514070.80 | $0.73 \pm 0.05$ | 0.38 |
| GIN | 281992.60 | $0.75 \pm 0.04$ | 0.15 |
| BS-KAGIN | 189140.10 | $0.76 \pm 0.05$ | 2.24 |
| RBF-KAGIN | 62553.50 | $0.76 \pm 0.03$ | 0.24 |

Table 14: Average time and parameter sizes for selected models, PROTEINS_full

| Model | Parameter count | Accuracy | time per epoch (s) |
|---|---|---|---|
| GCN | 14055.70 | $0.63 \pm 0.04$ | 0.09 |
| BS-KAGCN | 26241.00 | $0.76 \pm 0.03$ | 0.15 |
| RBF-KAGCN | 77950.00 | $0.74 \pm 0.04$ | 0.11 |
| GAT | 1217234.00 | $0.60 \pm 0.01$ | 0.14 |
| BS-KAGAT | 377421.60 | $0.73 \pm 0.02$ | 0.28 |
| RBF-KAGAT | 406871.00 | $0.72 \pm 0.04$ | 0.13 |
| GIN | 935245.00 | $0.71 \pm 0.03$ | 0.10 |
| BS-KAGIN | 206523.00 | $0.73 \pm 0.03$ | 0.36 |
| RBF-KAGIN | 76445.40 | $0.74 \pm 0.03$ | 0.14 |

Table 15: Average time and parameter sizes for selected models, ENZYMES

| Model | Parameter count | Accuracy | time per epoch (s) |
|---|---|---|---|
| GCN | 230071.20 | $0.21 \pm 0.04$ | 0.06 |
| BS-KAGCN | 102457.20 | $0.59 \pm 0.04$ | 0.12 |
| RBF-KAGCN | 105379.80 | $0.57 \pm 0.08$ | 0.08 |
| GAT | 2193344.40 | $0.18 \pm 0.01$ | 0.10 |
| BS-KAGAT | 734566.80 | $0.51 \pm 0.12$ | 0.23 |
| RBF-KAGAT | 292237.60 | $0.50 \pm 0.07$ | 0.09 |
| GIN | 778435.40 | $0.55 \pm 0.06$ | 0.07 |
| BS-KAGIN | 199432.90 | $0.44 \pm 0.08$ | 0.21 |
| RBF-KAGIN | 135783.50 | $0.45 \pm 0.09$ | 0.10 |

Table 16: Average time and parameter sizes for selected models, IMDB-BINARY

| Model | Parameter count | Accuracy | time per epoch (s) |
|---|---|---|---|
| GCN | 106247.80 | $0.68 \pm 0.04$ | 0.15 |
| BS-KAGCN | 34710.10 | $0.73 \pm 0.05$ | 0.19 |
| RBF-KAGCN | 40357.50 | $0.73 \pm 0.04$ | 0.17 |
| GAT | 1482555.60 | $0.67 \pm 0.04$ | 0.18 |
| BS-KAGAT | 391572.00 | $0.72 \pm 0.04$ | 0.25 |
| RBF-KAGAT | 308826.00 | $0.71 \pm 0.05$ | 0.18 |
| GIN | 769820.20 | $0.73 \pm 0.03$ | 0.16 |
| BS-KAGIN | 144101.00 | $0.73 \pm 0.04$ | 0.30 |
| RBF-KAGIN | 159608.70 | $0.70 \pm 0.04$ | 0.20 |

Table 17: Average time and parameter sizes for selected models, IMDB-MULTI

| Model | Parameter count | Accuracy | time per epoch (s) |
|---|---|---|---|
| GCN | 83223.60 | $0.49 \pm 0.04$ | 0.23 |
| BS-KAGCN | 50801.40 | $0.50 \pm 0.04$ | 0.29 |
| RBF-KAGCN | 25513.00 | $0.50 \pm 0.04$ | 0.25 |
| GAT | 838081.80 | $0.48 \pm 0.05$ | 0.25 |
| BS-KAGAT | 129868.00 | $0.50 \pm 0.04$ | 0.33 |
| RBF-KAGAT | 340916.00 | $0.48 \pm 0.04$ | 0.27 |
| GIN | 875383.20 | $0.50 \pm 0.04$ | 0.24 |
| BS-KAGIN | 195207.70 | $0.50 \pm 0.04$ | 0.38 |
| RBF-KAGIN | 40460.40 | $0.49 \pm 0.04$ | 0.26 |

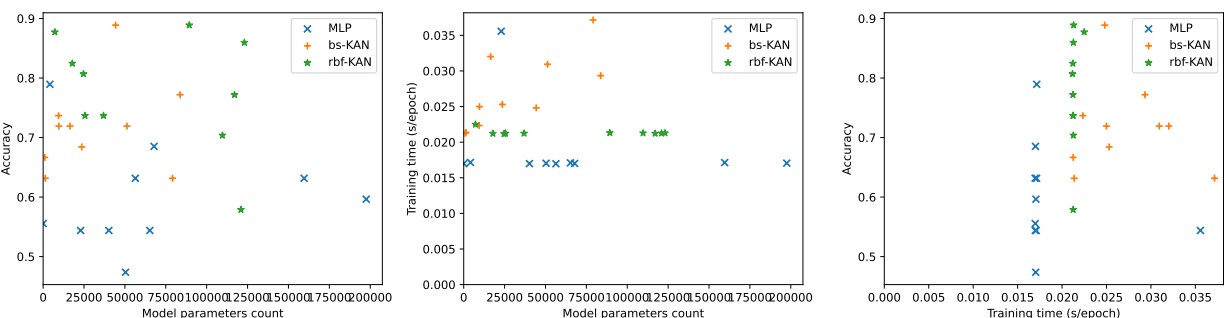

Figure 2: Parameters count/size/performance comparison, GCN, MUTAG

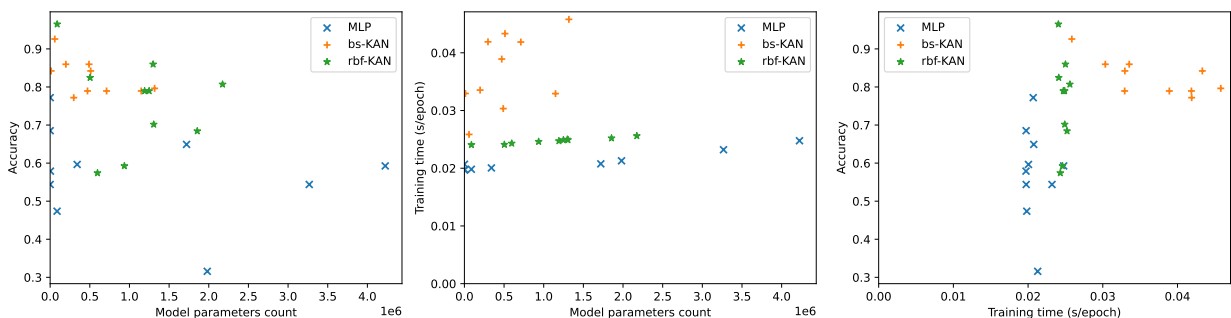

Figure 3: Parameters count/size/performance comparison, GAT, MUTAG

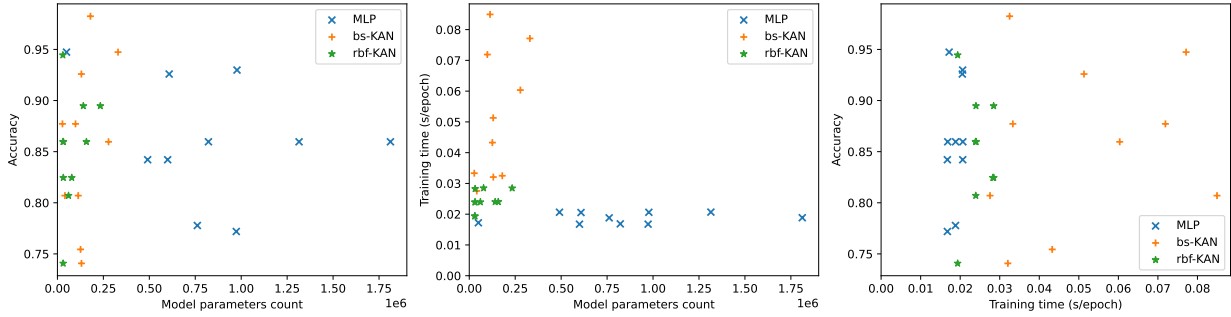

Figure 4: Parameters count/size/performance comparison, GIN, MUTAG

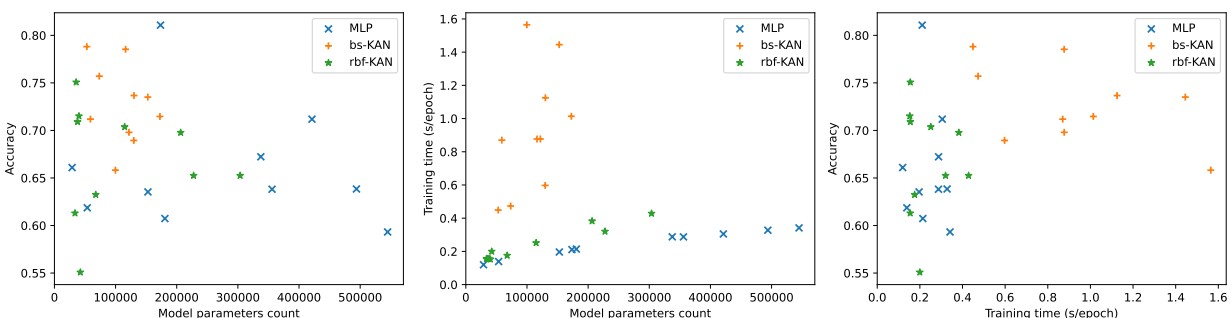

Figure 5: Parameters count/size/performance comparison, GCN, DD

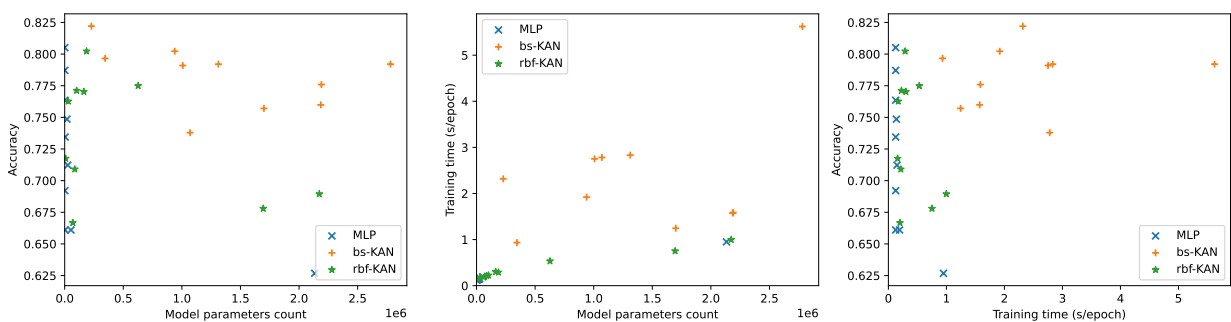

Figure 6: Parameters count/size/performance comparison, GAT, DD

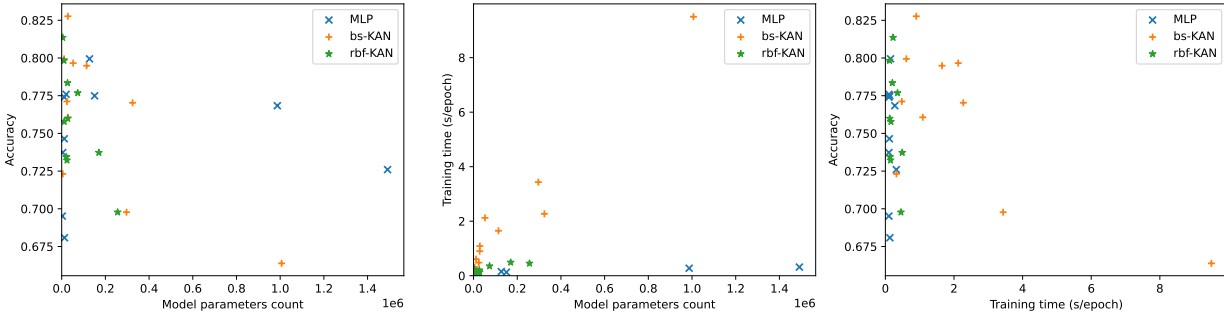

Figure 7: Parameters count/size/performance comparison, GIN, DD

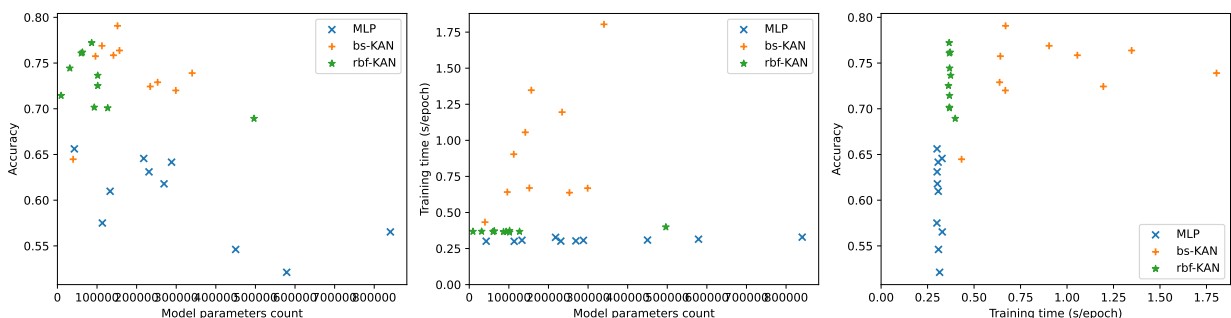

Figure 8: Parameters count/size/performance comparison, GCN, NCI1

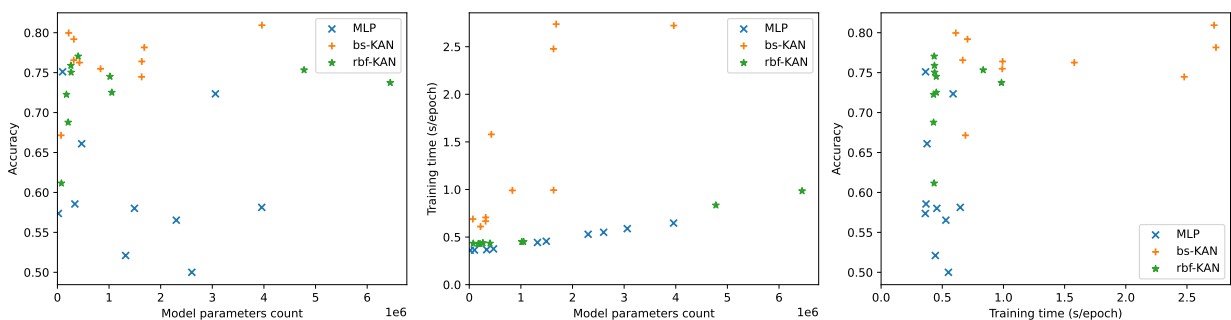

Figure 9: Parameters count/size/performance comparison, GAT, NCI1

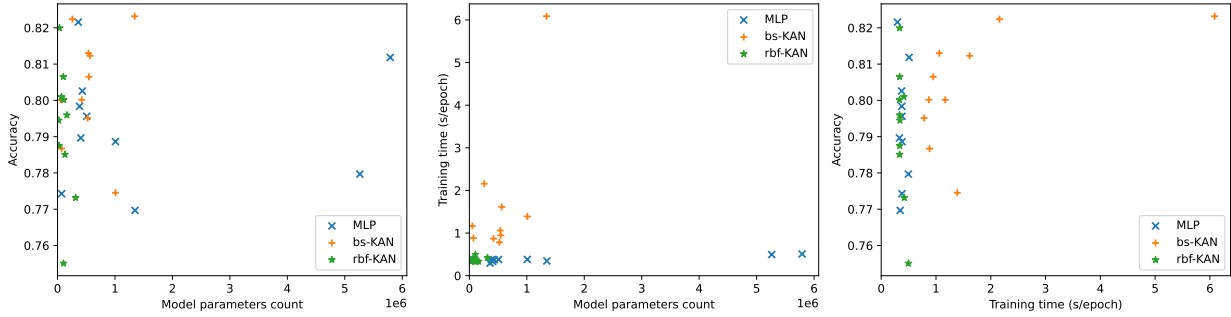

Figure 10: Parameters count/size/performance comparison, GIN, NCI1

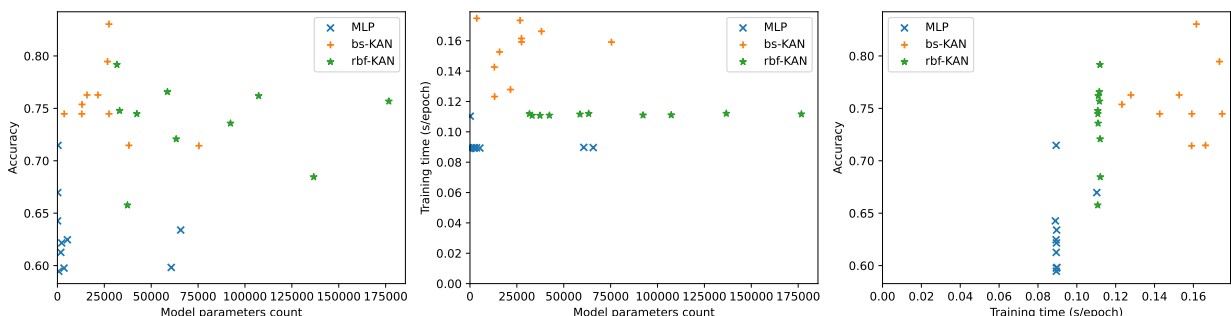

Figure 11: Parameters count/size/performance comparison, GCN, PROTEINS_full

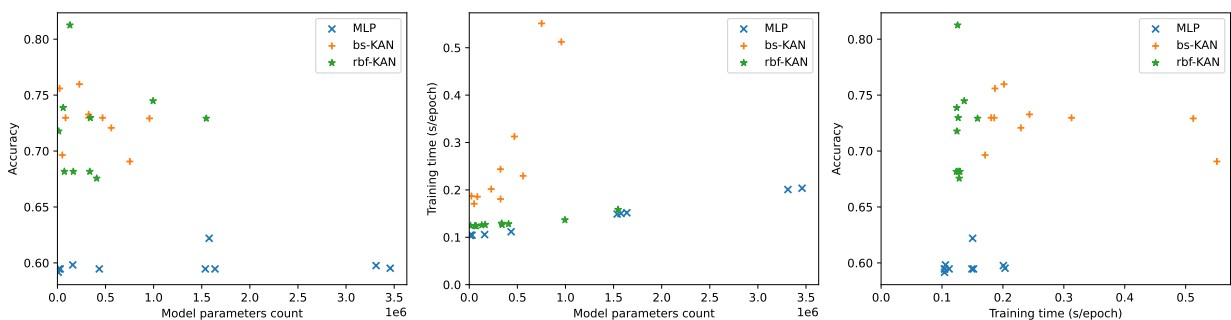

Figure 12: Parameters count/size/performance comparison, GAT, PROTEINS_full

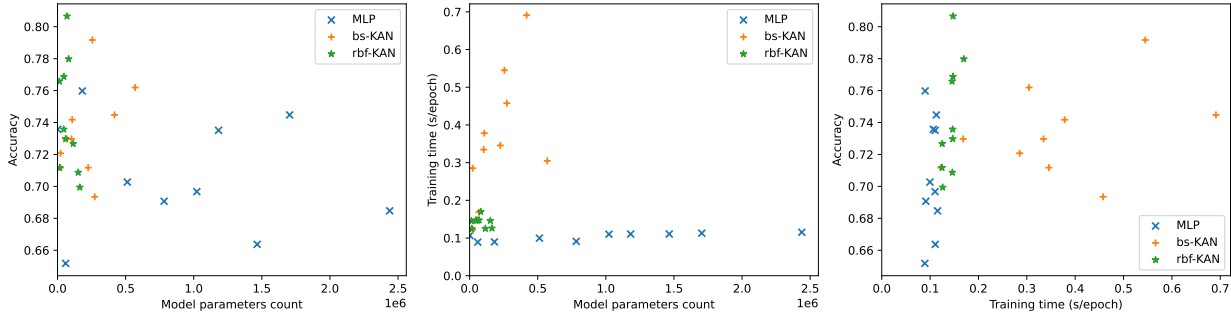

Figure 13: Parameters count/size/performance comparison, GIN, PROTEINS_full

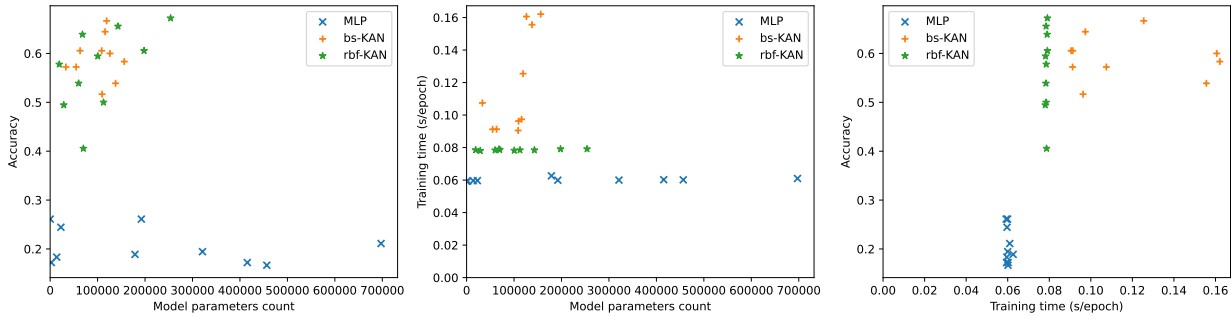

Figure 14: Parameters count/size/performance comparison, GCN, ENZYMES

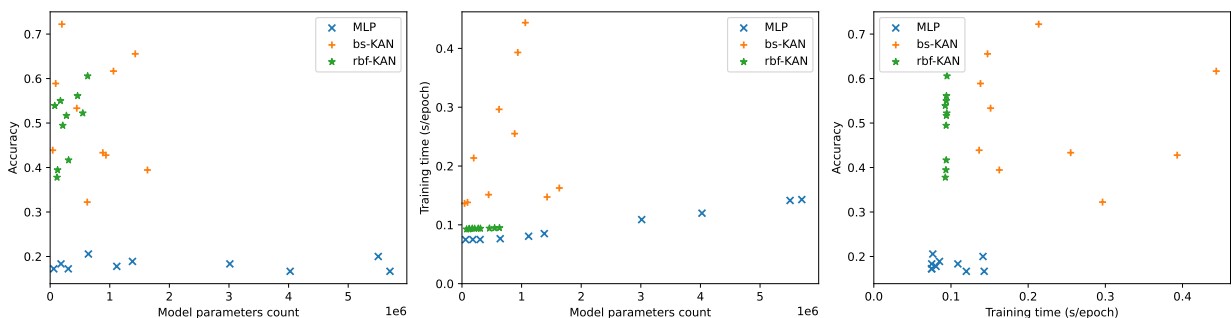

Figure 15: Parameters count/size/performance comparison, GAT, ENZYMES

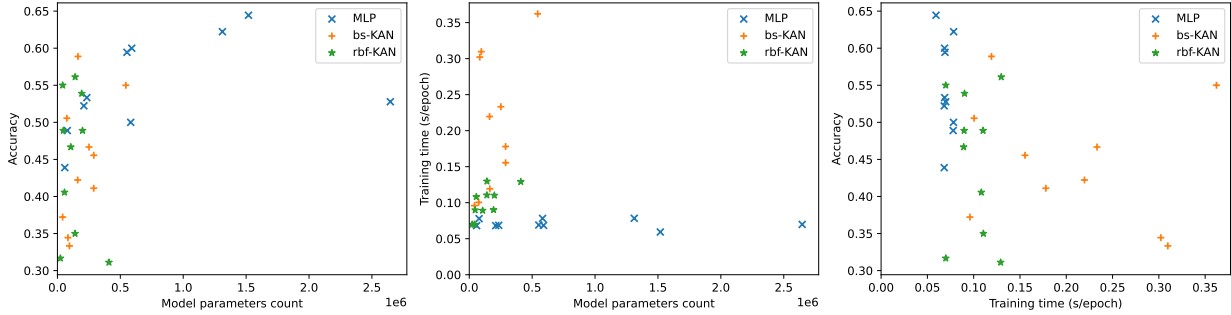

Figure 16: Parameters count/size/performance comparison, GIN, ENZYMES

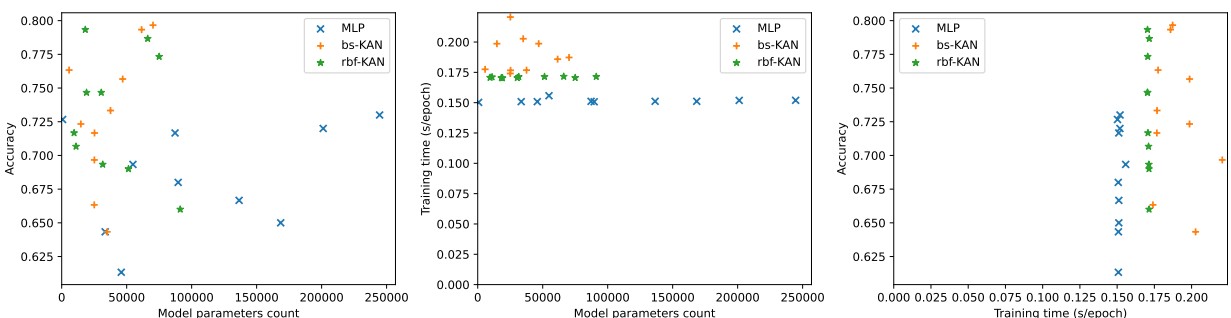

Figure 17: Parameters count/size/performance comparison, GCN, IMDB-BINARY

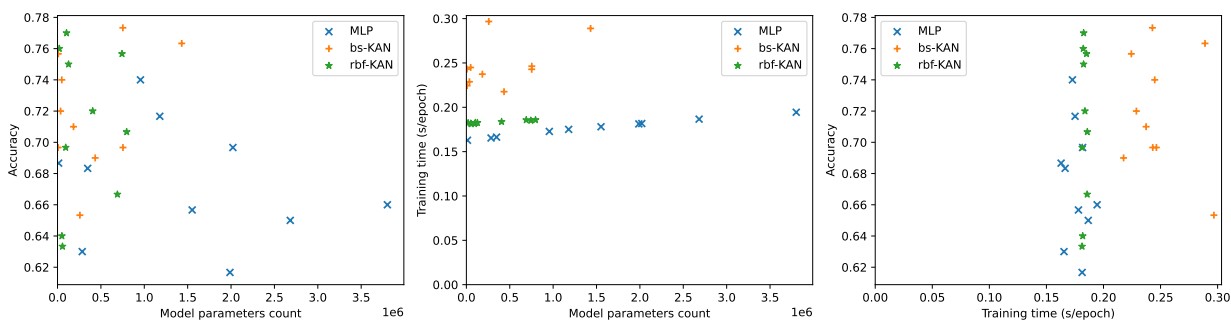

Figure 18: Parameters count/size/performance comparison, GAT, IMDB-BINARY

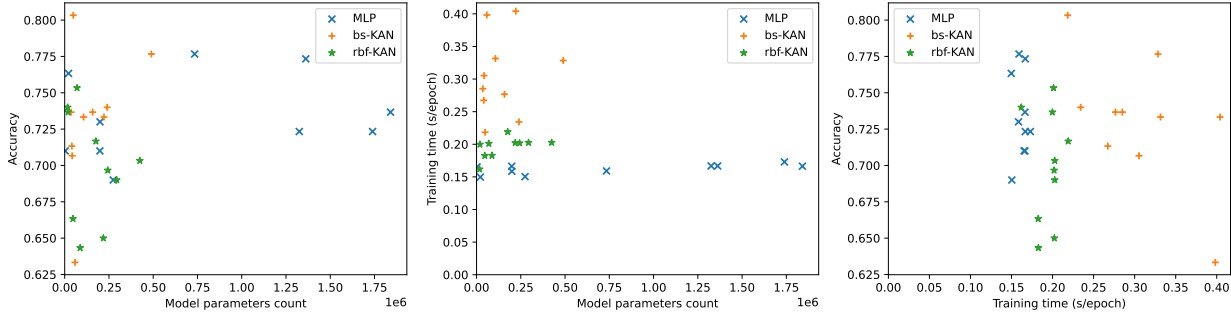

Figure 19: Parameters count/size/performance comparison, GIN, IMDB-BINARY

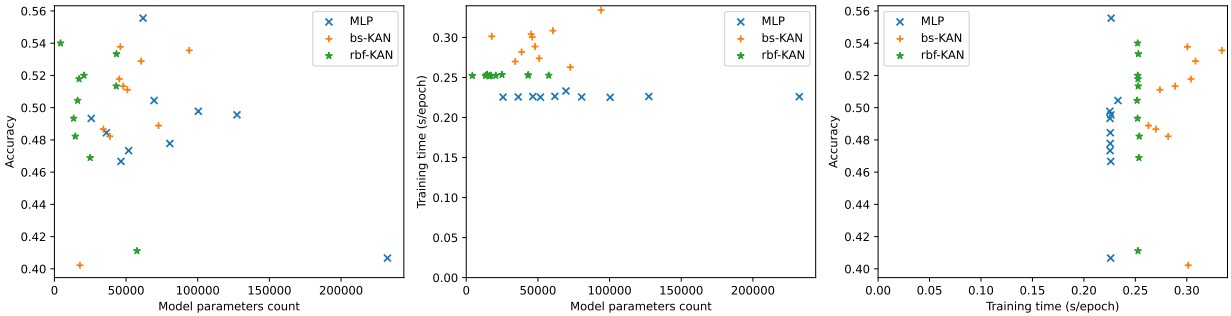

Figure 20: Parameters count/size/performance comparison, GCN, IMDB-MULTI

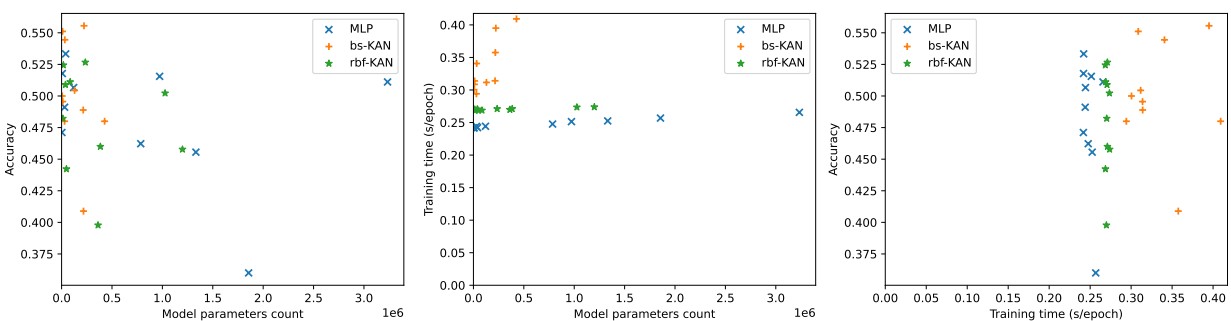

Figure 21: Parameters count/size/performance comparison, GAT, IMDB-MULTI

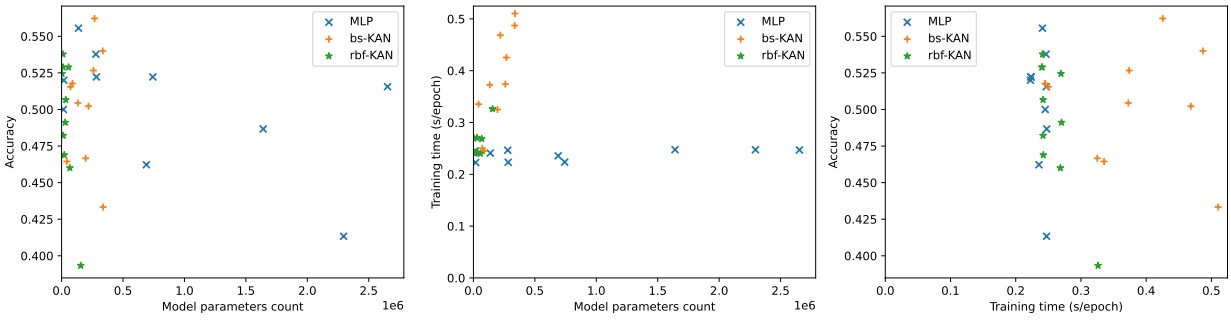

Figure 22: Parameters count/size/performance comparison, GIN, IMDB-MULTI

## C.1 Training Times - Graph Classification

We give in Table 18 the training time per epoch for different configurations of the `BS-KAGIN`, `RBF-KAGIN` and GIN models on a graph-level task. We observe that for a given number of message-passing layers and hidden dimension size, GIN is the fastest, followed by `RBF-KAGIN`. `BS-KAGIN` is noticeably slower than the others. It suffers doubly: its parameter count increases much quicker than both other models for same hyperparameters; but even for similar numbers of parameters, `BS-KAGIN` is by far the slowest. Spline order seems to have a larger impact on running time than grid size. This shows that RBFs do alleviate some of the computational bottlenecks of the B-splines from the original KAN paper. Table 19 in Appendix C.2 shows similar observations on a node-level task.

Table 18: Training time per epoch on NCI1 dataset. 5 message-passing layers. batch size set to 128.

| Architecture | Hidden Dimension | Hidden Layers | Grid Size | Spline Order | # Parameters | Training time (s/epoch) |
|---|---|---|---|---|---|---|
| GIN | 256 | 2 | NA | NA | 670722 | 0.303 |
| GIN | 256 | 4 | NA | NA | 1465346 | 0.390 |
| GIN | 512 | 2 | NA | NA | 2652162 | 0.541 |
| GIN | 512 | 4 | NA | NA | 5814274 | 0.951 |
| RBF-KAGIN | 256 | 1 | 1 | NA | 822872 | 0.386 |
| RBF-KAGIN | 256 | 1 | 4 | NA | 1639274 | 0.599 |
| RBF-KAGIN | 256 | 3 | 1 | NA | 3191408 | 0.783 |
| RBF-KAGIN | 256 | 3 | 4 | NA | 6367142 | 1.577 |
| RBF-KAGIN | 512 | 1 | 1 | NA | 3218520 | 0.807 |
| RBF-KAGIN | 512 | 1 | 4 | NA | 6424170 | 1.441 |
| RBF-KAGIN | 512 | 3 | 1 | NA | 12674160 | 2.198 |
| RBF-KAGIN | 512 | 3 | 4 | NA | 25317030 | 4.429 |
| BS-KAGIN | 256 | 1 | 1 | 1 | 1091072 | 0.446 |
| BS-KAGIN | 256 | 1 | 1 | 4 | 1907456 | 2.964 |
| BS-KAGIN | 256 | 1 | 4 | 1 | 1907456 | 0.720 |
| BS-KAGIN | 256 | 1 | 4 | 4 | 2723840 | 4.240 |
| BS-KAGIN | 256 | 3 | 1 | 1 | 4236800 | 0.937 |
| BS-KAGIN | 256 | 3 | 1 | 4 | 7412480 | 9.703 |
| BS-KAGIN | 256 | 3 | 4 | 1 | 7412480 | 1.963 |
| BS-KAGIN | 256 | 3 | 4 | 4 | 10588160 | 14.572 |

## C.2 Training Times - Node Classification

Table 19: Training time per epoch on the node-classification task (Cora dataset, 3 MP-layers).

| Architecture | Hidden Dimension | Hidden Layers | Grid Size | Spline Order | # Parameters | Training time (s/epoch) |
|---|---|---|---|---|---|---|
| GCN | 256 | NA | NA | NA | 378934 | 0.003 |
| GCN | 1024 | NA | NA | NA | 1485622 | 0.005 |
| BS-KAGCN | 256 | NA | 1 | 1 | 1514684 | 0.017 |
| BS-KAGCN | 256 | NA | 1 | 4 | 2650697 | 0.160 |
| BS-KAGCN | 256 | NA | 4 | 1 | 2650697 | 0.031 |
| BS-KAGCN | 256 | NA | 4 | 4 | 3786710 | 0.222 |
| BS-KAGCN | 512 | NA | 1 | 1 | 2989244 | 0.020 |
| BS-KAGCN | 512 | NA | 1 | 4 | 5231177 | 0.179 |
| BS-KAGCN | 512 | NA | 4 | 1 | 5231177 | 0.039 |
| BS-KAGCN | 512 | NA | 4 | 4 | 7473110 | 0.254 |
| RBF-KAGCN | 256 | NA | 1 | NA | 1142524 | 0.016 |
| RBF-KAGCN | 256 | NA | 4 | NA | 2278543 | 0.031 |
| RBF-KAGCN | 1024 | NA | 1 | NA | 4462588 | 0.029 |
| RBF-KAGCN | 1024 | NA | 4 | NA | 8916367 | 0.057 |
| GIN | 256 | 2 | NA | NA | 867335 | 0.007 |
| GIN | 256 | 4 | NA | NA | 1130503 | 0.008 |
| GIN | 1024 | 2 | NA | NA | 5042183 | 0.020 |
| GIN | 1024 | 4 | NA | NA | 9240583 | 0.033 |

| | | | | | | |
|---|---|---|---|---|---|---|
| BS-KAGIN | 256 | 1 | 1 | 1 | 1514684 | 0.017 |
| BS-KAGIN | 256 | 1 | 1 | 4 | 2650697 | 0.158 |
| BS-KAGIN | 256 | 1 | 4 | 1 | 2650697 | 0.032 |
| BS-KAGIN | 256 | 1 | 4 | 4 | 3786710 | 0.224 |
| BS-KAGIN | 256 | 3 | 1 | 1 | 3990528 | 0.025 |
| BS-KAGIN | 256 | 3 | 1 | 4 | 6983424 | 0.229 |
| BS-KAGIN | 256 | 3 | 4 | 1 | 6983424 | 0.051 |
| BS-KAGIN | 256 | 3 | 4 | 4 | 9976320 | 0.334 |
| BS-KAGIN | 512 | 1 | 1 | 1 | 2989244 | 0.021 |
| BS-KAGIN | 512 | 1 | 1 | 4 | 5231177 | 0.184 |
| BS-KAGIN | 512 | 1 | 4 | 1 | 5231177 | 0.041 |
| BS-KAGIN | 512 | 1 | 4 | 4 | 7473110 | 0.261 |
| BS-KAGIN | 512 | 3 | 1 | 1 | 10078208 | 0.046 |
| BS-KAGIN | 512 | 3 | 1 | 4 | 17636864 | 0.354 |
| BS-KAGIN | 512 | 3 | 4 | 1 | 17636864 | 0.089 |
| BS-KAGIN | 512 | 3 | 4 | 4 | 25195520 | 0.499 |
| RBF-KAGIN | 256 | 1 | 1 | NA | 1142524 | 0.017 |
| RBF-KAGIN | 256 | 1 | 4 | NA | 2278543 | 0.032 |
| RBF-KAGIN | 256 | 3 | 1 | NA | 3002487 | 0.023 |
| RBF-KAGIN | 256 | 3 | 4 | NA | 5995401 | 0.046 |
| RBF-KAGIN | 1024 | 1 | 1 | NA | 4462588 | 0.03 |
| RBF-KAGIN | 1024 | 1 | 4 | NA | 8916367 | 0.057 |
| RBF-KAGIN | 1024 | 3 | 1 | NA | 21429879 | 0.091 |
| RBF-KAGIN | 1024 | 3 | 4 | NA | 42838665 | 0.185 |

