# OpenReview forum: "KAGNNs: Kolmogorov-Arnold Networks meet Graph Learning"
_TMLR — Accepted by TMLR_

### Review · Reviewer_jhqf · 2024-12-19

**Summary Of Contributions:**

Graph neural networks perform two operations in each layer: feature transformation and feature aggregation. Traditionally, MLP shave been used for feature transformation. This paper replaces MLPs with recently proposed Kolmogorov-Arnold Networks (KANs) for feature transformation. It provides empirical evaluations on various graph related tasks including node classification and graph classification/regression. Its a great problem to study in the context of graph learning. However, this paper has limited novelty.

**Audience:**

Yes

**Claims And Evidence:**

Yes

**Requested Changes:**

It would be great if the authors can explore more on interpretability part as mentioned above, at least for some tasks.

Also, there is loose sentence in abstract " Our results indicate that KANs are on-par with or better than MLPs on all studied tasks,..." it should be in terms of graph learning, not generic.

**Strengths And Weaknesses:**

Although the experiments are extensive, I believe this work has limited novelty. Since KANs are known for its interpretability, in my opinion, this work should have focused on that aspect. For example, how the learned activations can explain the interactions between features as well as nodes. What kind of activations are prominent in social networks and what activations are prominent in molecular networks. The chosen activations are as in KAN paper, what about exploring domain specific activations in the context of graph learning tasks? Overall, without these insights, the paper just seems to be a accuray tables and benchmarks by replacing MLPs with KANs. I see that these questions have been mentioned in the future work section, but this should have been the focus of this paper in my opinion.

---

> ### Author Response · Authors · 2024-12-30
> **Answer - Reviewer jhqf**
>
> We would like to sincerely thank the reviewer for the thoughtful review.
>
> >Although the experiments are extensive, I believe this work has limited novelty. Since KANs are known for its interpretability, in my opinion, this work should have focused on that aspect. For example, how the learned activations can explain the interactions between features as well as nodes. What kind of activations are prominent in social networks and what activations are prominent in molecular networks. The chosen activations are as in KAN paper, what about exploring domain specific activations in the context of graph learning tasks? Overall, without these insights, the paper just seems to be a accuray tables and benchmarks by replacing MLPs with KANs. I see that these questions have been mentioned in the future work section, but this should have been the focus of this paper in my opinion.
>
> >It would be great if the authors can explore more on interpretability part as mentioned above, at least for some tasks.
>
>
> We understand the interest of the reviewer in the interpretability/explainability of KAN-based models. We are also interested in this direction. However, we believe that this falls out of the scope of this paper, and requires its own study. Indeed, our paper aims at benchmarking KAN-based models in graph learning settings, as without a strong empirical performance interpretability would be irrelevant. We therefore focused in this paper on showing that KAN-based models can be competitive with MLP-based models, motivating thus the need for future studies to explore the models' interpretability.
>
> Note also that according to TMLR's acceptance criteria (https://jmlr.org/tmlr/acceptance-criteria.html), novelty of the studied method is not a necessary criteria for acceptance of a paper. It is also our opinion that the reported experimental results shed light on the usefulness of KANs in the context of graph machine learning, and thus the criterion of interest is satisfied.
>
> >Also, there is loose sentence in abstract " Our results indicate that KANs are on-par with or better than MLPs on all studied tasks,..." it should be in terms of graph learning, not generic.
>
> We thank the reviewer for mentioning this sentence, which we fixed to avoid any ambiguity that we only consider the tasks tackled in this paper.

---

### Review · Reviewer_ywSt · 2024-12-20

**Summary Of Contributions:**

This paper proposes replacing Multi-Layer Perceptrons (MLPs) with Kolmogorov-Arnold Networks (KANs) as the update function in Graph Neural Networks (GNNs). The authors introduce two KAN-based GNN variants (KAGIN and KAGCN) and use two types of KANs (B-Splines and RBFs). Experiments are conducted on node and graph classification tasks, demonstrating that KAN-based GNNs perform competitively with or better than their MLP-based counterparts, particularly in regression-tasks.

**Audience:**

Yes

**Claims And Evidence:**

Yes

**Requested Changes:**

* (critical) Include experiments on link prediction.
* (critical) Add comparisons with more GNN architecture, at least, e.g., with Graph Attention Networks (GAT).
* (critical) Expand Table 9 and similar with model performance.

**Strengths And Weaknesses:**

Strengths:

* Novel use of KANs in GNNs
* Detailed experimental evaluation: The study conducts carries out an evaluation across multiple datasets for node and graph classification, providing empirical support for their claims.
* Clear insights into computational trade-offs: The paper discusses the advantages and disadvantages, providing practitioners with guidance for selecting the appropriate KAN variants.
* Practical relevance: The work has potential applications in tasks requiring smooth function approximations or interpretable models.

Weaknesses:

The strength of this work lies in its empirical evaluation. However, the evaluation is somewhat limited:

* Task scope:  The study focuses on node and graph classification tasks but omits link prediction, though it is an important application for GNNs.
* GNN architectures: The comparisons are limited to basic GNN variants (GIN and GCN), without benchmarking against more advanced models such as Graph Attention Networks (GAT).

These limitations reduce the generality of the findings but are straightforward to address.

---

> ### Author Response · Authors · 2024-12-30
> **Answer - Reviewer ywSt**
>
> We would like to thank the reviewer for the constructive comments and suggestions. Find below our response to your concerns.
>
> >Task scope: The study focuses on node and graph classification tasks but omits link prediction, though it is an important application for GNNs.
>
> >Requested Changes: (critical) Include experiments on link prediction.
>
>
> Following the reviewer's request, we have evaluated the KAN-based architectures in the task of link prediction. Specifically, we have experimented with the Cora and CiteSeer benchmark datasets and have added the results in subsection 4.2 of the revised manuscript. The results indicate that the KAN-based models perform on-par with MLP-based GNNs in the link prediction task.
>
>
> >GNN architectures: The comparisons are limited to basic GNN variants (GIN and GCN), without benchmarking against more advanced models such as Graph Attention Networks (GAT).
>
> >Requested Changes:
> >(critical) Add comparisons with more GNN architecture, at least, e.g., with Graph Attention Networks (GAT).
> >(critical) Expand Table 9 and similar with model performance.
>
>
> The reviewer is right in that the landscape of GNNs is not limited to GCN and GIN. In fact the literature of GNNs is large and there are now several well-established architectures (e.g., DGCNN, GAT, GraphSAGE, Graphormer).
>
> This paper does not aim at providing an exhaustive empirical analysis that considers a large number of GNN models. Instead, we focused on two models and evaluated those models in a variety of tasks (i.e., node classification, graph classification, graph regression) to provide insights into different scenarios. We chose the GIN and GCN models since these models are popular in different applications, while they both use fully-connected layers/MLPs to update the features of the nodes.
>
> While it would be interesting to evaluate the performance of other methods (e.g., GAT, GraphSAGE, or any more complex architecture), it would require implementing a KAN-based version of all of those models to compare each pair of architectures. While this is interesting work, we believe it warrants its own study, with a deeper analysis of the effect of KANs on each component of the models (e.g., attention mechanism).
>
> Finally, we do not agree with the reviewer that the GAT model is more advanced than GIN. In fact it is well-known that GIN is more expresive than GAT [1].
>
> [1] Xu, K., Hu, W., Leskovec, J., and Jegelka, S. "How powerful are graph neural networks?", In ICLR'19.

---

> > ### Comment · Reviewer_ywSt · 2025-01-01
> >
> > I would like to thank the authors for their response, manuscript update, and clarifications. However, two small points remain:
> >
> > 1. **Table 9 in the Original Manuscript**: I would like to gently follow up on my initial request concerning Table 9 in the original manuscript (now Table 17 in the revision). If the authors have decided not to update that table and instead moved it to the appendix, could they say so in their reply for clarity?
> >
> > 2. **GNN Models**: Although I understand the authors' focus on GIN and GCN, this paper is largely empirical and would benefit from demonstrating how KANs perform across a variety of models. If that is not feasible within the current scope, the authors could perhaps highlight this as a limitation or propose it as a direction for future work. Doing so will help readers understand the scope (and limits) of the work.
> >
> > Lastly:
> > > Finally, we do not agree with the reviewer that the GAT model is more advanced than GIN. In fact it is well-known that GIN is more expressive than GAT [1].
> >
> > I thank the authors for this clarification. Indeed, Xu et al. (2019) shows GIN is as expressive as the 1-WL isomorphism test, which is considered "maximally powerful" among standard message-passing GNNs. However, I would appreciate it if the authors could reference the specific part of Xu et al. that states (or implies) GIN is strictly more expressive than GAT. I emphasize that this request is purely to clarify this tangential point and does not affect my evaluation of the work.
> >
> > I thank the authors again for their efforts

---

> > > ### Author Response · Authors · 2025-01-02
> > >
> > > We thank the reviewer for their comment, and apologize for misunderstanding your request about Table 9 (now 17). We will do our best to adress them in this comment.
> > >
> > > **Table 9/17:** This table focuses on showing the effect of the different hyperparameters on computational overhead. While studying in detail the effect of each of those on the performance, those models were only trained for a small number of epochs, without finetuning the other hyperparameters or waiting for convergence. We thus cannot provide their performance. Nonetheless, section 5 provides a study relating the size and performance of the models that were selected by the optuna framework in the context of graph classification.
> > >
> > > **GNN Models:** We thank you for mentioning that our work's limitations were not made clear enough. We added mentions on the limitation of the studied models in the abstract, introduction and perspectives.
> > >
> > > **GAT vs GIN**: it is true that this claim requires clarification. While Xu et al shows that GIN is at least as expressive as GAT, it does not show strict domination. We thus offer the following sources which argue in the direction that GIN is indeed more expressive than GAT.
> > >
> > > "Will More Expressive Graph Neural Networks do Better on Generative Tasks?", Zou et al., Learning on Graph Conference LoG2023
> > >
> > > "How Attentive are Graph Attention Networks?", Brody et al., ICLR2022

---

### Review · Reviewer_KEvU · 2024-12-21

**Summary Of Contributions:**

This paper presents a comprehensive comparison between Multilayer Perceptrons (MLPs) and Kernel Attention Networks (KANs) within the context of Graph Neural Networks (GNNs). Specifically, the authors introduce modifications to the GCN and GIN architectures, replacing the original MLP layers with KANs to create KAGCN and KAGIN. They conduct extensive experiments on both node classification and graph classification tasks to demonstrate that KANs are more parameter-efficient and achieve better performance than MLPs.

**Audience:**

Yes

**Claims And Evidence:**

No

**Requested Changes:**

1. Elaborate on the motivation behind replacing MLPs with KANs, explicitly addressing how KANs improve expressiveness in ways relevant to the constraints of traditional GNNs (e.g., the 1-WL test).

2. Clearly articulate the contributions of the paper. Emphasize the unique aspects of KANs and their integration into GNNs beyond the evaluation of their performance.

3. Provide empirical evidence or illustrative examples to support the claim that KANs are more explainable than MLPs, as stated in Section 6. This could include case studies, attention heatmaps, or other interpretability analyses.

**Strengths And Weaknesses:**

Strengths:

1. The paper offers a thorough comparison between KANs and MLPs within GNN frameworks, shedding light on their respective advantages.

2. The manuscript is well-written, clearly structured, and easy to follow, which facilitates comprehension of the concepts and results.


Weaknesses:


1. The motivation for the study requires further clarification. The authors aim to enhance the expressiveness of GNNs by incorporating KANs. However, it is unclear how this aligns with the standard definition of GNN expressiveness, particularly the constraints imposed by the 1-WL test in basic message-passing GNNs. A detailed discussion of how KANs address these limitations or extend GNN expressiveness is necessary.

2. The paper’s main contribution is somewhat ambiguous. While the authors evaluate KAN-based GNNs against MLP-based GNNs, the specific novelty or advancement introduced by the KAN architecture is not explicitly emphasized. Clarifying the unique insights or techniques brought forth by this work would strengthen its impact.

3. In Section 6, the authors assert that KANs offer superior explainability compared to MLPs. However, no empirical evidence or qualitative examples are provided to substantiate this claim. Supporting this assertion with experiments, visualizations, or real-world examples would significantly enhance the credibility and relevance of this argument.

---

> ### Author Response · Authors · 2024-12-30
> **Answer - Reviewer KEvU**
>
> We wholeheartedly thank you for your valuable comments and their clear assessment.
>
> >The motivation for the study requires further clarification. The authors aim to enhance the expressiveness of GNNs by incorporating KANs. However, it is unclear how this aligns with the standard definition of GNN expressiveness, particularly the constraints imposed by the 1-WL test in basic message-passing GNNs. A detailed discussion of how KANs address these limitations or extend GNN expressiveness is necessary.
>
> >Requested Changes:
> >Elaborate on the motivation behind replacing MLPs with KANs, explicitly addressing how KANs improve expressiveness in ways relevant to the constraints of traditional GNNs (e.g., the 1-WL test)."
>
>
> We understand that there is a confusion there. We do not pretend to enhance the expressiveness of GNNs. In fact, we claim to work within the traditional message passing framework described in section 2.2, whose upper bound for representativity is indeed the 1-WL test. As such, the upper bound for expressivity is reached for a $\phi$ (in Equation (1)) which is a universal approximator. GIN uses an MLP to reach this upper bound, we propose a KAN (also a universal approximator) as an alternative. As a consequence, as we claim in 3.1, our KAGIN model is as expressive as GIN.
>
> Of course, this is in theory, since in practice, both MLPs and KANs are not actually universal approximators, while in a given dataset it is not known whether proper weights will be learned due to the limitations of the training algorithm. Therefore, while KANs **cannot** enhance the expressive power of message-passing GNNs and make them more powerful than 1-WL, they could potentially impact the training dynamics and allow the models to easier reach their full potential. Our experiments aim to investigate this aspect.
>
>
> >"The paper’s main contribution is somewhat ambiguous. While the authors evaluate KAN-based GNNs against MLP-based GNNs, the specific novelty or advancement introduced by the KAN architecture is not explicitly emphasized. Clarifying the unique insights or techniques brought forth by this work would strengthen its impact.
>
> >Requested Changes:
> >Clearly articulate the contributions of the paper. Emphasize the unique aspects of KANs and their integration into GNNs beyond the evaluation of their performance.
>
> This paper aims at benchmarking KANs as feature update functions in GNNs, from a performance point of view, and establish that they are relevant for the community to be further studied. We indeed showed that the performance of the KAN-based models (out-)matches that of MLP-based models, and that this does not necessarily come at the cost of a high computational overhead. In fact, we even illustrate a tradeoff between a slower, but arguably more accurate implementation and a faster one, which matches MLP in terms of time complexity and performance, albeit with a much lower number of parameters.
>
> Note that according to TMLR's acceptance criteria (https://jmlr.org/tmlr/acceptance-criteria.html), novelty of the studied method is not a necessary criteria for acceptance of a paper. It is also our opinion that the reported experimental results shed light on the usefulness of KANs in the context of graph machine learning, and thus the criterion of interest is satisfied.
>
>
> >In Section 6, the authors assert that KANs offer superior explainability compared to MLPs. However, no empirical evidence or qualitative examples are provided to substantiate this claim. Supporting this assertion with experiments, visualizations, or real-world examples would significantly enhance the credibility and relevance of this argument.
>
> >Requested Changes:
> >Provide empirical evidence or illustrative examples to support the claim that KANs are more explainable than MLPs, as stated in Section 6. This could include case studies, attention heatmaps, or other interpretability analyses.
>
> We should stress that this paper focuses exclusively on benchmarking KAN models in the context of GNNs. In section 6, we provides perspectives and also discuss some future directions for research. Our claim that KANs offer superior explainability compared to MLPs is motivated by the results reported in the original KAN paper. We understand that this has led to confusion and we have updated the paper accordingly, adding in particular a source from the original paper's author focusing on the interpretability of KANs.
>
> We believe that we first need to ensure that KAN models are relevant and competitive with MLPs in the context of graph learning, and then investigate the explainablility of those models. This paper focuses on the former and we expect that its positive findings will ignite interest in the community to theoretically study the explainability of those models.

---

### Decision · Action_Editor_BVoJ · 2025-02-03

**Recommendation:** Accept with minor revision

**Comment:**

The paper proposes modifications to the GCN and GIN architectures by substituting the original MLP layers with KANs, resulting in the KAGCN and KAGIN models. Extensive experiments on node and graph classification tasks demonstrate that KANs are more parameter-efficient and outperform baseline methods. It is interesting and new to add KANs to GNNs and improve their performance. The experiments and the paper writing are overall good. It is necessary to further improve the work by adding elaborated motivations behind replacing MLPs with KANs, providing link prediction results, and exploring more architectures like GAT.

**Audience:**

The paper will attract a good number of TMLR's audience in graph learning research.

**Claims And Evidence:**

Yes while needs some minor revisions.